# A Systematic Exploration of Satellite Radar Coherence Methods for Rapid Landslide Detection

Katy Burrows[1], Richard J. Walters[1], David Milledge[2], and Alexander L. Densmore[3]

[1]COMET, Department of Earth Sciences, Durham University
[2]School of Engineering, Newcastle University
[2]Department of Geography, Durham University

**Correspondence:** Katy Burrows (katy.a.burrows@durham.ac.uk)

**Abstract.** Emergency responders require information on the distribution of triggered landslides within two weeks of an earthquake or storm. Useable satellite radar imagery is acquired within days of any such event worldwide. Recently, several landslide detection methods that use these data have been developed, but testing of these methods has been limited in each case to a single event and satellite sensor. Here we systematically test five methods using ALOS-2 and Sentinel-1 data across four triggering earthquakes. The best performing method was dependent on the satellite sensor. For three of our four case study events, an initial ALOS-2 image was acquired within 2 weeks, and with these data, co-event coherence loss (CECL) is the best performing method. Using a single post-event Sentinel-1 image, the best-performing method was the boxcar-sibling method. We also present three new methods which incorporate a second post-event image. While the waiting time for this second post-event image is disadvantageous for emergency response, these methods perform more consistently and on average 10% better across event and sensor type than the boxcar-sibling and CECL methods. Thus, our results demonstrate that useful landslide density information can be generated on the timescale of emergency response, and allow us to make recommendations on the best method based on the availability and latency of post-event radar data.

## 1 Introduction

Information on the spatial distribution of earthquake- or rainfall-triggered landslides needs to be generated as quickly as possible in order to be useful for emergency response efforts, ideally within two weeks of an event (Inter-Agency Standing Committee, 2015; Williams et al., 2018). This information is commonly generated from analysis of optical satellite imagery (e.g., Bessette-Kirton et al., 2019; Kargel et al., 2016). However, relying solely on optical satellite imagery in landslide assessment is problematic, as the mapping process can be significantly delayed by cloud cover (Robinson et al., 2019). For example, following the 2015 $M_w$ 7.8 Gorkha, Nepal earthquake, almost no cloud-free optical imagery was acquired over the region of most intense landsliding for a full week following the earthquake (Williams et al., 2018), and Robinson et al. (2019) have

demonstrated that this delay could have been much longer had the earthquake occurred during Nepal's monsoon season, with some areas unlikely to be successfully imaged at all between June and September.

When optical imagery is not available, empirical models based on factors such as the topographic slope and measurements or predictions of earthquake-induced shaking or rainfall data are used to predict the likely location and intensity of triggered landsliding (e.g. Kirschbaum and Stanley, 2018; Kritikos et al., 2015; Nowicki Jessee et al., 2018). The outputs of these models have a comparatively low spatial resolution (around 1 km$^2$ in the case of Nowicki Jessee et al. (2018)), but can be used to provide an overview of the most severely impacted areas. While these models can be generated within hours of an earthquake or rainfall event, they do not always perform well. For example, the USGS ground failure model of Nowicki Jessee et al. (2018) under-predicted the landsliding triggered by the $M_w$ 6.9 2018 Lombok, Indonesia earthquake sequence (Ferrario, 2019) and multiple empirical models over-predicted the spatial extent of landslides triggered by the 2016 $M_w$ 7.8 Kaikoura, New Zealand earthquake (Allstadt et al., 2018). The reliability of global empirical rainfall-triggered landslide susceptibility maps such as that of Kirschbaum and Stanley (2018) is dependent on the type of landslides, on the rainfall dataset used and on the intensity of the triggering rainfall event (Jia et al., 2020; Kirschbaum and Stanley, 2018). Similarly, the accuracy of empirical models for earthquakes is strongly dependent on the quality of the shaking data or model. Earthquake-induced shaking information is typically updated several times following an event, and the details of the shaking dataset that is used can have a strong effect on the modelled landslide spatial distribution and impacts (Allstadt et al., 2018; Robinson et al., 2017). For example, empirical models generated immediately following the Gorkha earthquake failed to capture the spatial pattern of triggered landslides (Robinson et al., 2017). This spatial information is critically important for use in emergency response coordination, so this limitation is a significant disadvantage when applying such empirical models.

Synthetic aperture radar (SAR) satellite imagery presents a means of generating landslide information in all weather conditions as radar is able to penetrate cloud. For landslide studies, SAR is most commonly used to measure the downslope velocity of slow-moving landslides (e.g. Aslan et al., 2020; Bonì et al., 2018; Dai et al., 2016; Handwerger et al., 2019; Hu et al., 2019; Reyes-Carmona et al., 2020; Solari et al., 2020). However, SAR can also be used to detect modifications to the Earth's surface and it has been demonstrated that radar methods can be used to automatically detect wind damage to forests (Rüetschi et al., 2019), flooding (Martinis et al., 2015) and urban damage following earthquakes, typhoons and wildfires (Fielding et al., 2005; NASA, 2018; Yun et al., 2015).

Recently, there have been several attempts to develop similar SAR-based change detection methods for rapid landslide mapping, based on SAR amplitude (Konishi and Suga, 2018, 2019; Mondini et al., 2019), coherence (Burrows et al., 2019; Olen and Bookhagen, 2018; Yun et al., 2015), or some combination of these (Aimaiti et al., 2019; Jung and Yun, 2019), or based on polarimetric SAR methods (e.g. Yamaguchi et al., 2019). However, with the exception of Mondini et al. (2019) who used a global selection of landslides, these studies are generally tested on a single landslide event and use a single radar sensor. For example, Aimaiti et al. (2019); Konishi and Suga (2019); Jung and Yun (2019); Yamaguchi et al. (2019) tested their methods using ALOS-2 imagery of the 2018 Hokkaido earthquake. If such methods are to be applied in future events, wider testing is needed. This would allow us to establish whether different methods work equally well for different events, and to determine the best method to use with data from a given SAR sensor and within a given time window.

To address this need, we carried out a systematic statistical comparison of the performance of five radar-based methods of landslide detection, using imagery acquired by the Sentinel-1 and ALOS-2 PALSAR-2 sensors spanning four case study landslide-triggering earthquakes. We chose to test on earthquakes rather than rainfall events for two reasons. First, it can be assumed that the majority of landslides occurred concurrently with or very shortly after the shaking, and this information on landslide timing simplifies the validation of the methods. Second, radar imagery is more likely to be acquired immediately after an earthquake as part of emergency tasking of satellite acquisitions, as these data are commonly used to measure earthquake-related ground deformation. We tested on four large ($M_w > 6.6$) events: the 2015 Gorkha, Nepal earthquake, the 2018 Hokkaido, Japan earthquake, and two earthquakes of the 2018 Lombok, Indonesia sequence (Fig. 1). All of these events triggered thousands of landslides, which have been mapped using optical satellite imagery (Ferrario, 2019; Roback et al., 2018; Zhang et al., 2019). We assessed the ability of each method and radar dataset to predict these validation data, and demonstrate the wide applicability of SAR coherence methods to landslide detection, making recommendations for which method is most suitable depending on the type of SAR data that is available and the timing of data acquisition.

## 2 Satellite radar coherence for change detection

A SAR system works by illuminating the Earth's surface with microwave radiation and measuring the amplitude and phase of the returned signal. In the interferometric SAR (InSAR) technique, the difference in phase between two images acquired over the same area at different times can be used to map the change in distance between the ground and the satellite. SAR amplitude is the strength of the backscattered signal and is partially dependent on the material at the ground surface: its orientation relative to the satellite, its roughness, and its dielectric properties.

When using an interferogram to map ground deformation, it is important that the signals recorded at a given location in the two SAR images are correlated, as decorrelation will result in high-frequency noise. In order to assess this and to identify noisy pixels, the coherence $\gamma$ is estimated for every pixel from the similarity in the two SAR images in amplitude and phase difference, for a small ensemble of $n$ pixels (Eq. 1, Just and Bamler, 1994):

$$\gamma = \frac{\frac{1}{n}\sum_{i=1}^{n} A_i \cdot \overline{B_i}}{\sqrt{\frac{1}{n}(\sum_{i=1}^{n} A_i \cdot \overline{A_i} \sum_{i=1}^{n} B_i \cdot \overline{B_i})}} \tag{1}$$

$A_i$ and $B_i$ are complex representations of the phase and amplitude of each pixel $i$ within the ensemble, with the complex conjugate shown by the overline. The ensemble is chosen so that the pixels used in the calculation are expected to be similar. In a "boxcar" method, it is assumed that pixels immediately adjacent to and centred on the target pixel are similar to it (e.g. Hanssen, 2001; Yun et al., 2015). In a "sibling" method an assessment is carried out for every pixel to identify pixels that are statistically similar to it. For example, the sibling method of Spaans and Hooper (2016) identifies pixels that have similar amplitude behaviour through time.

Coherence can be decomposed into 3 components, with the total coherence dependent on their product (Eq. 2, Zebker and Villasenor, 1992):

$$\gamma_{total} = \gamma_{temporal} \cdot \gamma_{spatial} \cdot \gamma_{thermal} \qquad (2)$$

Here we are interested in temporal coherence $\gamma_{temporal}$, as decorrelation of this component reflects changes in the physical properties of the Earth's surface between image acquisitions. The spatial coherence, $\gamma_{spatial}$, is dependent on the geometric properties of the satellite acquiring the image and the ground surface. Decorrelation of the spatial component of coherence $\gamma_{spatial}$ is the result of small changes in satellite viewing geometry between acquisitions and can be stronger in areas of steep topography, as it is dependent on incidence angle. This decorrelation is particularly sensitive to the SAR image pair's perpendicular baseline (the distance between the locations at which the satellite acquired the two SAR images measured perpendicular to both the flight and look directions). When the perpendicular baseline of the image pair used to form an interferogram is sufficiently small, this spatial component will be small compared to any temporal decorrelation (Zebker and Villasenor, 1992). For modern satellites, this will be the case most of the time. We removed distorted pixels, which were likely to be more strongly affected by decorrelation of $\gamma_{spatial}$, from our analysis in Section 3.4. Decorrelation of the thermal coherence $\gamma_{thermal}$ was assumed to be insignificant, following Zebker and Villasenor (1992).

## 3 Data and methods

### 3.1 SAR data

In this study, radar imagery was used from two satellite systems. Sentinel-1 uses C-band radar (wavelength $\sim$ 5.6 cm), whilst ALOS-2 PALSAR-2 uses lower frequency L-band radar (wavelength $\sim$ 24 cm). The difference in wavelength between the two systems means that, in forested areas, L-band radar penetrates further into the canopy than C-band. The shorter wavelength of C-band radar means it is sensitive to surface modifications on a smaller spatial scale. For example, in a forest, C-band radar data may detect change in the location or orientation of leaves, while L-band is sensitive instead to changes in the location and orientation of branches. L-band InSAR is often more useful in vegetated areas, as its deeper penetration allows it to retain higher coherence in the absence of major vegetation changes (Zebker and Villasenor, 1992).

Radar systems acquire data at an oblique angle to the vertical on near-polar ascending and descending orbital tracks (referred to here as a and d), which are acquired on different dates. The satellite look direction is perpendicular to the orbit direction. The data used in this study were acquired at an angle of between 31.4° and 43.8° to vertical. This oblique acquisition angle means that on a given track, some hillsides will be more favourably oriented to the sensor than others, and so information from ascending and descending tracks can be combined to obtain more complete coverage of an event. As it is impossible to calculate a combined coherence surface using data from two tracks, and in some cases imagery will only be acquired on one track within two weeks of an event, here we considered each track separately. We will refer to tracks according to their sensor, track number, and orbit direction, for example, A018d (ALOS-2, track 18, descending orbit).

## 3.2 Case studies

We used four case study events: the 2015 $M_w$ 7.8 Gorkha, Nepal earthquake (Fig. 1a, d); the $M_w$ 6.6 2018 Hokkaido, Japan earthquake (Fig. 1b, e); and two $M_w$ 6.8 and 6.9 earthquakes from the 2018 Lombok, Indonesia sequence (Fig. 1c, f). These four events have several traits in common which made them suitable for this study. First, they were all large earthquakes, triggering thousands of landslides, making them of interest from an emergency response perspective. This also meant that the earthquakes and associated landslides had previously been investigated, and inventories of triggered landslides had been compiled from optical satellite imagery, enabling direct testing of the radar methods against these independent datasets (Ferrario, 2019; Roback et al., 2018; Zhang et al., 2019). Second, while the vegetation types differ between the three regions, the presence of dense vegetation across each region meant that we could expect landslide and non-landslide pixels to have a similar first-order signal in the radar data across the events. Third, ascending and descending track Sentinel-1 data and at least one track of high resolution ALOS-2 data (acquired in stripmap mode at a resolution of 3 - 10 m) were available for all case study areas. Finally, the type of landslides triggered by our four case study earthquakes were typical of landslides triggered by earthquakes (Keefer, 1984). The majority of ground failures in the four earthquakes were slides. In Nepal, ground failures were primarily a mixture of slides and falls, with the exception of a large debris avalanche in the Langtang Valley. For all four earthquakes, failure surfaces were at shallow depths in most cases with a small number of exceptions (Collins and Jibson, 2015; Ferrario, 2019; Yamagishi and Yamazaki, 2018).

The $M_w$ 7.8 2015 Gorkha, Nepal earthquake (Fig. 1a) occurred on 25 April 2015 and triggered around 25,000 landslides over an area hundreds of kilometres wide, as mapped by Roback et al. (2018). Sentinel-1 imagery was acquired on tracks S019d and S085a. ALOS-2 data on track A157a were divided into subtracks with acquisitions on different dates, shown as separate polygons on Fig. 1a, which will be referred to as East (E), Central (C) and West (W).

The $M_w$ 6.6 2018 Hokkaido, Japan earthquake (Fig. 1b) occurred on 5 September 2018. Two inventories have been published for this event: one containing 7,837 landslides (Wang et al., 2019) and one containing 5,265 (Zhang et al., 2019). Neither provided information on the mapping extent so we assumed that this could be approximated by the convex hull of the data locations. As the inventory of Zhang et al. (2019) has the largest convex hull, we used this inventory for validation of the radar methods. Both descending and ascending ALOS-2 data were available for this event, with a higher spatial and temporal frequency than for the other events. The earthquake occurred the day after Typhoon Jebi passed over Hokkaido, and so this case study was also an opportunity to test landslide detection methods following a rainfall event, with the advantage that because the typhoon and earthquake occurred one day apart, and aerial imagery of the triggered landslides was acquired immediately afterwards, we know more precisely when the landslides occurred (Yamagishi and Yamazaki, 2018). This is important if SAR methods are to be used in the future for mapping storm-triggered landslides because factors such as wind damage and the water content of the soil are known to affect SAR coherence and amplitude (Rüetschi et al., 2019; Scott et al., 2017).

The 2018 Lombok earthquake sequence comprised 4 earthquakes with $M_w$ > 6: $M_w$ 6.4 on 28 July; $M_w$ 6.8 on 5 August; and $M_w$ 6.3 and 6.9 on 19 August. Ferrario (2019) generated two landslide inventories for this sequence. Although cloud-free imagery was not available across the whole affected area following the earthquake on 28 July, no landslides were visible in the

areas that could be mapped. Ferrario (2019) thus presented their first inventory of 4,823 landslides triggered following the 5 August earthquake, and a second inventory of 9,319 landslides which had been triggered by the end of the sequence. We refer to the 5 August inventory as Lombok-1 (Fig. 1c) and the 19 August inventory as Lombok-2 (Fig. 1c, inset).

There are several key ways in which the events differed. First, the triggered landslides had very different spatial patterns, with the Gorkha earthquake triggering landslides across an area spanning hundreds of km, while the Lombok and Hokkaido earthquakes triggered landslides across only a few km. It can be seen in Fig. 1 that much denser landsliding was triggered by the Hokkaido earthquake than by the other events. Second, the sizes and shapes of the landslides were very different between events: the Lombok earthquakes triggered a large number of small landslides with a median landslide area of 460 $m^2$ for Lombok-1 and 580 $m^2$ for Lombok-2, compared to equivalent median areas of 4,350 $m^2$ for Hokkaido and 1,070 $m^2$ for Nepal (measured from the inventories of Ferrario, 2019; Roback et al., 2018; Zhang et al., 2019). Third, the events occurred under different weather conditions: the Hokkaido earthquake followed months of heavy rain and occurred one day after Typhoon Jebi, while the Gorkha and Lombok earthquakes occurred during the dry season. Finally, the topographic relief in the three case study areas varied significantly. In Nepal the majority of landslides occurred on slopes of over 40° (Roback et al., 2018). For the Hokkaido event, the proportion of slopes > 40° was very low and so the majority of landslides occurred on much shallower slopes (Wang et al., 2019), with Lombok lying between these two extremes. This is highly relevant to the application of SAR coherence methods to landslide detection. Steep slopes can lead to distortion of the radar image, and coherence is also dependent on the geometry of the hillslope and radar sensor. Therefore, we might expect that, as hillslopes in Hokkaido and Lombok are shallower than in Nepal, landslide detection using SAR may be more successful in these areas. These differences between the four events made them ideal for testing the wider applicability of SAR-coherence-based landslide detection methods in vegetated areas. If a method is to be widely applied in the future, we need to be confident that its performance is consistent across differing events and settings.

### 3.3 Landslide detection methods

We tested two existing methods: the co-event coherence loss (CECL) method of Yun et al. (2015) and the boxcar-sibling method of Burrows et al. (2019). Each of these existing methods uses a single post-event SAR image. We also present three new methods that incorporated a second post-event image: the post-event coherence increase (PECI); the sum of the coherence increase and decrease ($\Delta C\_sum$); and the maximum of coherence increase or decrease ($\Delta C\_max$). A "boxcar" coherence estimate is used for CECL, PECI, $\Delta C\_sum$ and $\Delta C\_max$.

#### 3.3.1 Co-event coherence loss (CECL)

The coherence loss between a pre-event interferogram and a co-event interferogram can be used to detect physical changes to the ground surface associated with an earthquake, such as surface rupture, building damage and landslides (Fielding et al., 2005; Washaya et al., 2018; Yun et al., 2015). This method has been applied by the NASA Advanced Rapid Imaging and Analysis (ARIA) project for use in urban damage mapping and identifies "damaged" pixels as those where coherence has decreased in the co-event map relative to the pre-event map. First the pre-event coherence map is adjusted so that it has the same

coherence frequency distribution as the co-event coherence map using exact histogram matching (Coltuc et al., 2006). This process accounts for different levels of bulk temporal decorrelation in the pre-event and co-event interferograms. It assumes only a small fraction of the pixels are affected by landslides so that the landslide signal is not removed from the co-event interferogram. The pre-event surface is then subtracted from the co-event surface and pixels whose coherence has decreased are flagged as damaged (Yun et al., 2015). Although this method was developed for detecting damage to buildings, Yun et al. (2015) tested it on the 2015 Gorkha earthquake and noted that landslides in the Langtang Valley corresponded spatially to areas of coherence decrease in the ARIA surface. Coherence decrease between pre-event and co-event interferograms has since been used as an input in the landslide detection methods of Aimaiti et al. (2019) and Jung and Yun (2019) applied to the 2018 Hokkaido earthquake.

### 3.3.2   The boxcar-sibling method (Bx-S)

Rather than relying on the coherence change through time, the Bx-S method of Burrows et al. (2019) uses the difference between two alternative co-event spatial methods of coherence calculation as a landslide classification surface. The four other methods tested here all use a traditional "boxcar" coherence, in which the coherence of a pixel is estimated from the similarity in phase change of the pixels immediately adjacent to it. When a sibling-based method is used to estimate coherence, the coherence of a pixel becomes dependent on "siblings" that are not immediately adjacent to it but that are expected to behave similarly. In the method of Spaans and Hooper (2016) used here, an ensemble of siblings is selected for every pixel that have similar amplitude behaviour in a time series of pre-event imagery. For a landslide pixel, this means that its coherence is calculated from a more dispersed ensemble of pixels than with a traditional boxcar coherence estimate and so proportionally less of the ensemble will also lie within the landslide. A sibling-based coherence surface is therefore relatively insensitive to landslides, and landslides can be identified as those whose co-event boxcar coherence is lower than their co-event sibling coherence (see Burrows et al. (2019) for detail).

### 3.3.3   Post-event coherence increase (PECI)

The coherence decrease caused in a co-event interferogram by a landslide is a temporary effect, assuming that the landslide stops moving following the earthquake. Therefore in a post-event interferogram (calculated from two post-event images) coherence should be higher for landslide pixels than in a co-event interferogram. As landslides expose bare rock or soil, which is likely to have higher coherence than vegetated areas, this co-event to post-event increase may actually be larger than the pre-event to co-event decrease used in the CECL method to measure landslides, making the signal easier to detect. This is particularly the case when using C-band SAR, which experiences more decorrelation than L-band in vegetated regions. Applying the same histogram-matching step as in the CECL method, we propose this co-event to post-event coherence increase as a new potential landslide detection method.

### 3.3.4 Sum of coherence increase and decrease ($\Delta$C_sum)

As landslides are expected to exhibit both a decrease in co-event coherence and an increase in post-event coherence, we summed the absolute magnitudes of these changes to form a new landslide classification surface. As in the CECL method, the pre-event and post-event coherence maps were histogram-matched to the co-event coherence map, removing bulk changes in coherence between the three maps. This method is equivalent to CECL + PECI.

### 3.3.5 Maximum of coherence increase or decrease ($\Delta$C_max)

For every pixel, we took whichever was largest of the pre-event to co-event coherence loss (CECL) and the co-event to post-event increase (PECI). This method is similar to the $\Delta$C_sum method but uses whichever has the strongest signal. The relative signal strength of the CECL or PECI methods will vary by location, for example according to the vegetation type. This method takes whichever of the two methods has a stronger signal for any given location.

### 3.4 Data processing

SAR data were processed using GAMMA, with the LiCSAR processing software used for Sentinel-1 (Li et al., 2016). The data were multilooked by a factor of 5 in range and 1 in azimuth (Sentinel-1) or by 5 in both range and azimuth (ALOS-2) to improve the signal to noise ratio. See Table 1 for information on the data resolution and pixel size at various stages of the processing. For geometric coregistration, we used the 1-arcsecond Shuttle Radar Topography Mission (SRTM) digital elevation model (Farr et al., 2007).

The boxcar coherence estimate used in all methods was calculated using a $3 \times 3$ pixel moving window (Table 1). The sibling-based coherence estimate used for the Bx-S method was calculated using the RapidSAR algorithm of Spaans and Hooper (2016). Siblings were calculated based on all pre-event images shown in Fig. 1 (a minimum of 6 images). For every pixel, between 15 and 50 siblings were identified within an $81 \times 81$ pixel window based on their amplitude and amplitude variability (window sizes in Table 1). Unfortunately, insufficient pre-event ALOS-2 data were available to carry out this calculation in the 235 case of the 2015 Gorkha, earthquake and so the Bx-S method could not be tested for this case.

In most cases, we used the co-event, pre-event, and post-event coherences with the shortest temporal baseline; however for the inventory provided by Ferrario (2019) for the landslides triggered by the Lombok earthquake on 19 August, we used a co-event image that spans this earthquake and the 5 August earthquake, as the landslide inventory contains landslides triggered by both events. After calculating each landslide classification surface from the methods described in Section 3.3, we used 240 the GAMMA software to convert the surfaces to a geographic coordinate system. This process, involving reprojection and interpolation, results in uniform pixel size (20 m $\times$ 22 m). We then normalised the values of each surface by the theoretical maximum and minimum value that could be obtained from each method, resulting in a set of classification surfaces with values between 0 and 1, with 1 most likely to be a landslide.

Before statistical testing, we removed distorted pixels. The oblique angle at which SAR imagery is acquired meant that 245 some pixels were distorted by topography and were badly imaged by the SAR system, experiencing shadow, foreshortening,

or layover (Franceschetti et al., 1994) and decorrelation of $\gamma_{spatial}$ (Section 3.1). Following Burrows et al. (2019), we masked these according to the area in geographic coordinates that contributes to the pixel in radar geometry, removing pixels with an area of 0 and those where this area was over six times larger than their multilooked pixel spacing in radar coordinates (see Table 1).

We carried out statistical testing of the results at two resolutions: first, at the initial resolution of the processed radar data in geographical coordinates (20 m × 22 m), with the vector landslide inventories of Ferrario (2019); Roback et al. (2018); Zhang et al. (2019) rasterized at this resolution; and second, at an aggregated resolution of 200 m × 220 m, calculated by amalgamating 10 × 10 grids of the 20 m × 22 m pixels. Aggregated classifier pixels were given the mean value of the unmasked pixels in the 10 × 10 grid. If over 95% of an aggregate pixel was made up of masked pixels, the aggregate pixel was masked. This

high threshold of 95% was chosen to minimise the loss of spatial coverage due to the masks. Varying the threshold between 95% and 5% had little difference in terms of the number of pixels used in the analysis in Hokkaido and Lombok (<5%), but in Nepal, where more pixels were masked due to distortion on steep slopes, decreasing the threshold to 5% resulted in a loss of coverage of around 40% on S085a. Altering this threshold made very little difference to the results presented in Section 4.1.

    In this study, we did not attempt to map SAR classification surface values directly to landslide areal density values, as this

has not been attempted in previous studies (e.g. Aimaiti et al., 2019; Burrows et al., 2019; Jung and Yun, 2019; Yun et al., 2015) and may not be possible due to differences in viewing geometry, land cover and, particularly with the ALOS-2 data used here, differences in temporal baseline between events. Thus, a binary ground truth was preferable, which we generated by assigning aggregate pixels as "landslide" if they were composed of over 25% landslide by area according to the rasterized landslide inventories we used for verification. We explore the effect that this choice of a 25% threshold to define a landslide

pixel has on our results in the supplementary material, but varying this threshold between 1% and 50% was found to have little effect on the relative performance of the different classifiers.

    The aggregation process was done for several reasons. First, the boxcar coherence estimation has the effect of blurring neighbouring pixels, so that the minimum size of an object we could resolve was dependent on the size of the 3 × 3 boxcar (see Table 1). In all cases this was larger than the 20 m × 22 m pixel spacing in geographic coordinates. Second, it has already

been demonstrated that SAR coherence-based methods perform better at lower resolutions (Burrows et al., 2019). Third, when comparing landslide inventories, for example those of Zhang et al. (2019) and Wang et al. (2019) for the Hokkaido earthquake, there is often some variation in the exact shape and location of individual polygons. We expect this effect to be decreased when inventories are downsampled. We therefore chose a resolution that was larger than the boxcar window but that resulted in enough mapped landslide and non-landslide aggregate pixels for analysis. This is similar to the resolution of other landslide

products generated for emergency response (e.g. Bessette-Kirton et al., 2019; Nowicki Jessee et al., 2018)

### 3.5   ROC analysis

We used the receiver operating characteristic (ROC) area under the curve (AUC) to evaluate and compare the classification ability of each surface. For each classification surface, a threshold was set for which no pixels were classified as landslides and then incrementally decreased until it reached a value where all pixels were classified as landslides. At each incremental

threshold, the false positive rate (the ratio of false positives to real non-landslide pixels) was plotted against the true positive rate (the ratio of true positives to real landslide pixels) to form an ROC curve. The area under this curve is equal to the probability that if a landslide pixel and a non-landslide pixel were randomly selected from the dataset, the classifier would rank them correctly (Hanley and McNeil, 1982). For a randomly generated surface with no classification ability, the AUC = 0.5. For a perfect classifier, the AUC = 1.0.

On all SAR tracks, there are many more non-landslide than landslide pixels. It has been suggested that for such imbalanced data, precision-recall curves can better represent classification ability than ROC AUC (Saito and Rehmsmeier, 2015). Here, we chose to use ROC analysis since precision-recall curves do not allow comparison between datasets with different proportions of landslide and non-landslide pixels and therefore between different earthquakes and SAR tracks. However, when considering the relative performance of classifiers for each track independently, we found the same conclusions could be drawn from precision-recall curves as from ROC curves. A recreation of Figure 2a using precision-recall rather than ROC AUC values can be found in Supplementary Information.

# 4 Results

## 4.1 Results at 200 m × 220 m resolution

Figure 2 shows the ROC AUC values for each classification method described in Section 3.3 and each track of radar data shown in Fig. 1. The cells are coloured so that the best performing classification surfaces are shown in green and the weakest in red. A classifier that performs well for all events and both sensors should therefore appear as a green row. The eastern and western tracks of S157a and data for Lombok-2 on track A129a have been omitted, as the waiting times for the first post-event image were very long (77, 63 and 139 days respectively). This long time window resulted in widespread co-event coherence loss that adversely affected classifier performance. Such a long waiting time would make it extremely unlikely that these data could be used in emergency response and the poor performance is unlikely to be representative of classifier behaviour when using more timely imagery.

The methods are grouped by the number of post-event images that are required, and the waiting time in days for this imagery for each event and SAR track is also given. As the primary scientific use for post-event ALOS-2 imagery is to form a co-event interferogram, only one post-event image was acquired immediately following the Lombok-1 and Nepal earthquakes, with the waiting time for the second post-event image being considerably longer. We still include these results as the waiting time for L-band radar data is likely to decrease in the future with the planned NISAR satellite constellation, which is expected to launch in 2022, acquiring data globally with a 12 day repeat time (Sharma, 2019).

If we consider only data acquired within the two week emergency response window (Inter-Agency Standing Committee, 2015), the highest AUC in Hokkaido was initially 0.58 on day 0 using the Bx-S method on S046d, rising to 0.89 after 1 day with data from track A018d and the CECL method. In Nepal, the first radar data were acquired on day 4 on track S019d, from which the highest AUC of 0.74 was obtained using the Bx-S method. On day 7, the first ALOS-2 image was acquired on A157a, and more accurate information could have been generated using CECL (AUC 0.81), although this ALOS-2 scene

covers a smaller area than the Sentinel-1 data (Fig. 1). In Lombok, the 6-day Sentinel-1 acquisition repeat time meant that a large volume of data was available within two weeks of these events. The best data and method for Lombok-1 evolved as follows: CECL, S032d (day 0, AUC 0.56); CECL or Bx-S, S156a (day 3, AUC 0.64); PECI, S032d (day 6, AUC 0.69); ΔC_max, S156a (day 9, AUC 0.72); CECL, A129a (day 13, AUC 0.88). The corresponding classification surfaces for this evolution are plotted in Fig. 3. For Lombok-2, the first Sentinel-1 image was acquired on day 1, track S156a, and had an AUC of 0.67 using the Bx-S method, which could be slightly improved upon using the second post-event image that was acquired on day 7, using PECI (AUC 0.68). No ALOS-2 data was available within 2 weeks of this earthquake. Therefore in most cases, initially only Sentinel-1 data are available and the best option is the Bx-S method. This can then be improved upon when the first ALOS-2 image becomes available using CECL. For Lombok-1 and Lombok-2, where two post-event Sentinel-1 images became available before the first ALOS-2 image, incorporating these data also improved on the accuracy of the result.

With a single L-band post-event image from ALOS-2, CECL was the best-performing landslide classification surface in all cases. An improvement was seen when an additional post-event image was acquired, and methods requiring this image were used, for the case of Lombok and Nepal, but not for Hokkaido. When a single C-band post-event image was used, the Bx-S method outperformed CECL for Hokkaido, Gorkha and Lombok-2 and had a similar performance for Lombok-1. The addition of a second post-event image and adoption of methods that used this image showed an improvement in Hokkaido and Lombok-1, but not in Nepal. However, when considering Fig. 2a as a whole, methods that used a second post-event image were both better performing and more consistent across event and sensor type. When looking across all three events, the best option in terms of AUC was to use the ΔC_sum method with L-band imagery. When grouped by event and radar look direction and ranked according to AUC, Δ C_sum using ALOS-2 was ranked 3rd out of 10 classification surfaces for descending imagery over Hokkaido, 1/9 for ascending imagery over Nepal and 2/10 for ascending imagery over Lombok-1. A comparison cannot be made for ascending track data in Hokkaido, as A116a and S085a had different look directions (westwards and eastwards respectively).

Figure 4 shows the landslide classification surfaces calculated using the ΔC_sum method and ALOS-2 data of each event, alongside the aggregated validation landslide data. In Figure 4b, d, f, cells made up of <1% landslide by area are masked. In order to recreate this in the radar surface, it was necessary to threshold and plot only pixels that were most likely to be landslides based on their classifier value. Here we applied a threshold such that the number of pixels plotted in panels (a, c, e) is the same as the number in panels (b, d, f), respectively. These threshold values were similar, but not identical, and we expect that more case study sites would be required to determine a more general threshold for application in future events. However, in each case, the spatial pattern of landsliding in (b, d, f) was recreated using the radar surfaces (a, c, e), which would allow the worst affected areas to be identified for emergency response even without strict definition of this threshold.

## 4.2 Variation across event and sensor type

Our results show significant variation across event and sensor type. L-band radar outperformed C-band in the majority of cases, but for some methods, we observed additional effects causing differences in performance.

For methods relying on a co-event vs. pre-event coherence decrease (i.e. CECL, $\Delta C\_sum$, $\Delta C\_max$), landslides associated with Lombok-2 were more difficult to predict than Lombok-1 (Fig. 2). This is likely to be due to the co-event acquisition time window, which was 6 days for the first inventory, but had to be increased to 18 days on track S156a and 24 days on S032d in order to span both earthquakes. This will have caused temporal decorrelation of vegetated non-landslide pixels, resulting in

a smaller difference between landslide and non-landslide co-event coherence and making it more difficult for the classifier to distinguish between these. The same effect was seen in Nepal, where 12 day interferograms were used for S019d, but 24 day pre-event and co-event interferograms were used for S085a.

Generally, the Bx-S method was more consistent across sensor type than the CECL method. The CECL method performed better than the Bx-S method with ALOS-2 data, but worse with Sentinel-1. There are several possible reasons for this. First,

the longer wavelength of L-band SAR meant that it was able to maintain a higher coherence in the pre-event interferogram, so that the coherence difference for a landslide pixel in a pre-event and co-event interferogram was larger, resulting in better performance from CECL. Second, in the Bx-S method, siblings are identified using pre-event imagery. As Sentinel-1 imagery is acquired every 12 days, more images were available for this calculation and were acquired over a shorter time period. This allows less time for pixels to be altered by changes to the ground surface, meaning that, for non-landslide pixels, a pixel and

its siblings are likely to be more similar. In this way, the siblings selected by RapidSAR for Sentinel-1 imagery may have been of a higher quality than those for ALOS-2, giving a more reliable coherence estimate.

### 4.3 Results at 20 m × 22 m resolution

While the main aim of this study was not to map individual landslides, we also show results at a 20 m x 22 m scale (the resolution of the classification surfaces in geographic coordinates), with pixels that are sufficiently small to resolve individual

landslides. Our results show that SAR methods were less successful at this resolution than when downsampled, with AUC on average 16 % lower using Sentinel-1 data and 11 % lower using ALOS-2 data. Using Sentinel-1 data, AUC values were low, ranging from 0.49 (CECL, Lombok-2) to 0.61 (PECI, Hokkaido). From this we conclude that mapping individual landslides with Sentinel-1 data was not possible using the methods tested here, which supports similar preliminary findings by Burrows et al. (2019).

Classification surfaces using ALOS-2 are more promising, with AUC up to 0.80 ($\Delta C\_max$, Lombok-1). As in Section 4.1, $\Delta C\_sum$ performed best, having the highest AUC on tracks A116a and A157a, and an AUC of only 0.01 less than the best performing classification surface on tracks A018d and A129a. Figure 6 shows small regions of this surface overlain with landslide polygons for each event. While the landslides generally coincide spatially with high classifier values, it is clear that it would not have been possible in most cases to map the landslide polygons using the radar data. Some large landslides (Fig.

6b, d) contain pixels with high classifier values. However, in (d), there are also areas of mapped landslide that do not have high classifier values, and in (a) and (c) the landslides are too small. Therefore we conclude that, while radar coherence methods show promise in individual landslide detection, it was not possible using the methods tested here, and there is likely to be a minimum detectable landslide size.

## 5 Discussion

We have demonstrated that SAR data is widely applicable to landslide detection in vegetated areas within the timeframe of the emergency response effort. For example, within the two week limit suggested by Inter-Agency Standing Committee (2015) and Williams et al. (2018), it would have been possible to generate triggered landslide density information using the CECL method with ALOS-2 data with an ROC AUC of 0.81 in Nepal, 0.89 in Hokkaido and 0.88 in Lombok-1 (Fig. 2). In this section, we first consider the applications of the SAR methods to future events, and then turn to potential sources of errors. Finally we discuss future work that would allow wider application of SAR methods to landslide detection.

### 5.1 Application

#### 5.1.1 Landslide density estimation

We found that radar methods were better suited to the production of landslide density maps than to the identification of individual landslides. The aggregated resolution of 200 m x 220 m that we used here was not high enough to identify individual landslides. However it was higher than the resolution of most empirical landslide susceptibility models designed for rapid response (Allstadt et al., 2018; Nowicki Jessee et al., 2018) and landslide maps generated from optical satellite imagery for use in aid efforts by Bessette-Kirton et al. (2019) following Hurricane Maria, Puerto Rico in 2017, and by Williams et al. (2018) following the 2015 Gorkha earthquake. SAR data seem best suited to producing products at this spatial scale. For example, Bessette-Kirton et al. (2019) produced a grid of 2 x 2 km pixels and assigned them as "high landslide density" (>25 landslides), "low landslide density" (1-25 landslides) or "no landslides". This was published a month after the hurricane using optical imagery acquired between 6 and 18 days after the event. As radar data are available within a few days of an earthquake, equivalent products could easily be generated from radar within this timescale.

Landslide density maps can be combined with data on population density in order to estimate exposure, as in Nowicki Jessee et al. (2018). They can be used alongside maps of roads to identify transport routes that are likely to be blocked, and maps of river networks in order to identify areas where a landslide may have temporarily dammed a river, posing a risk of flash flooding when the dam collapses (Robinson et al., 2018). Information of this kind may guide aerial assessments of landslides such as that by Collins and Jibson (2015), which was carried out 32-36 days after the 2015 Gorkha earthquake to identify possible landslide dams.

#### 5.1.2 Recommendations on data and methods

We showed in Section 4 that, in the majority of cases, landslide classification surfaces generated using L-band data outperformed those generated from C-band data. Currently, the main source of L-band data is the ALOS-2 satellite system. This system has a 14 day repeat time and one of its objectives is disaster response. Therefore, in most cases, a post-seismic image will be made available within two weeks of an earthquake, with the main purpose of producing a co-event interferogram that can be used to measure ground displacement and infer fault geometry. However, as this interferogram requires only one post-

event image, the waiting time for a second image may be several months, as was the case following the Nepal and Lombok earthquakes. Therefore in an emergency response situation, it is likely that only one L-band image will be available within the required timeline, and so CECL would be the best available classification surface. L-band data are also less likely to be acquired following a rainfall event, even if this event triggers many landslides, as ground displacement maps and therefore interferograms are not needed. In the future, however, the availability of L-band SAR is likely to increase with the planned NISAR and ROSE-L satellite constellations. NISAR, a joint NASA-ISRO mission planned to launch in 2022, will acquire L-band data continuously with a 12-day repeat time over all landmasses globally (Sharma, 2019), while the ESA ROSE-L satellite, whose launch date is planned in 2026, will have 6-day global repeat coverage, and 3-day in Europe (Pierdicca et al., 2019).

While Sentinel-1 data yield generally lower AUC values (Figs. 2a, 5), the system acquires data continuously over global tectonic belts with a 12 day repeat time, with all data made freely available. Image acquisitions on ascending and descending tracks are offset in time so that the waiting time for a single post-event track is always less than 6-12 days. Over Europe, Sentinel-1 data are acquired at twice this frequency, meaning that the first post-event image should be available within 3 days. As the data are regularly acquired, they will be available for rainfall events as well as earthquakes. In some cases, we may have two post-event Sentinel-1 images before the first ALOS-2 image becomes available. It is more difficult to make a definitive recommendation for the best method to use in this situation, as performance varies significantly between events. With one post-event image, the Bx-S method is the best performing classifier with Sentinel-1, and in Nepal remains the best classifier even after additional post-event Sentinel-1 images have been acquired. With two post-event images, the best performing classifier in 2 out of 8 cases is either $\Delta C\_sum$ or $\Delta C\_max$. However in cases where the presence of vegetation means that pre-event coherence and non-landslide co-event coherence are very low, the best option is to use PECI (e.g., Lombok-1 S032d, Lombok-2, Hokkaido). Inspection of the pre-event coherence should therefore be carried out first to select which method to use.

## 5.2 Sources of incorrect classifications

### 5.2.1 Building damage

As the CECL method of Yun et al. (2015) was originally designed to detect urban damage, it is unsurprising that damaged buildings cause false positives in SAR coherence-based methods for landslide detection. Large scale signals, such as the town of Atsuma in Hokkaido, could be removed using a land cover map, but masking landslides in built-up areas is clearly disadvantageous as this is where they would do the most damage. Additionally, buildings located outside of large towns may not be included in such maps, but damage to these would still result in false positives. These false positives will occur using all the methods in this study. They are particularly likely to have negatively affected the AUC values for A116a in Hokkaido, as this track had a larger proportion of non-landslide pixels lying in built up areas.

### 5.2.2 Wind damage

In Hokkaido, we observed some large false-positive patches in CECL, PECI, $\Delta C\_sum$ and $\Delta C\_max$ classification surfaces using ALOS-2 data and PECI, $\Delta C\_sum$ and $\Delta C\_max$ surfaces using Sentinel-1. These correspond spatially to areas of ever-green needleleaf forest in the Japanese Aerospace Exploration Agency (JAXA) high resolution land use and land cover map of Japan (JAXA, 2018). Areas with this forest cover, which we have mapped using Sentinel-2 imagery, are outlined in white on Fig. 6b.

We hypothesise that this type of vegetation may have been damaged by wind during Typhoon Jebi, which passed over the area the day before the earthquake, causing a coherence decrease that is similar to that caused by landsliding. This type of forest has a comparatively high coherence in both pre-event and post-event L-band interferograms, giving it a strong signal in CECL, PECI, $\Delta C\_sum$ and $\Delta C\_max$. This effect may also explain why the low coherence area extends beyond the area affected by landslides (an effect visible in our data, and also observed by Fujiwara et al. (2019)). Unfortunately, none of the SAR data in this study were acquired between the typhoon and the earthquake, which would have allowed separation of the two events. However, these forest patches have no signal in amplitude-based methods, which has been observed in the study of Fransson et al. (2010) of wind-damaged forest at this resolution. This lack of an amplitude signal may allow these false positives to be removed through a combination of amplitude and coherence-based classification surfaces. Additionally, we observed that these patches of forest are generally much larger than the landslides and so could perhaps be removed if object- rather than pixel-based classification were used.

### 5.2.3 Snow

Snow is a potential source of error when using SAR in landslide detection. There is a strong difference in SAR backscattering properties between wet snow, dry snow and no snow (Koskinen et al., 1997). As such, decreased coherence can be caused by snow melt, drift or fall between image acquisitions. Examination of Sentinel-2 imagery shows there was no snow cover at the time of the Hokkaido earthquake, and snow is not likely in Lombok due to the high temperatures. We do not have exact snow cover data for Nepal at the time of the earthquake, but we expect that in April approximately 1/5 of the country's total area would have been covered in snow (ICIMOD, 2013). Therefore at high altitudes, it is reasonable to assume that some false positives may have been caused by snow, particularly on track A129a due to the long time interval between image acquisitions.

### 5.2.4 Rivers

The Bx-S method is effectively a spatial filter on co-event coherence, removing signals that cover a large area. Therefore small, low coherence objects will be identified as landslides. This includes rivers, as demonstrated by Spaans and Hooper (2016). As the low coherence caused by a river is not temporary, this should have less of an effect on the other coherence methods tested here. However, changes in position or water level are likely to result in false positives in all coherence-based classifiers. A variety of methods could be used to identify and remove rivers from our analysis (including using a pre-event Bx-S surface). However, since areas where landslides and rivers intersect are particularly hazardous due to the potential for landslide dams and

associated flash flooding, we did not mask rivers in this study. We suggest that any product based on SAR coherence supplied to emergency response coordinators should have rivers overlaid. This would both mask false positives due to rivers and allow identification of locations where rivers pass through areas of intense landsliding.

 ### 5.2.5 Landslide density

When using the Bx-S method in Hokkaido, we found low AUC values with both L-band and C-band SAR. Visual comparison of the boxcar and sibling coherence surfaces shows that the area affected by landslides had a low coherence in both, rather than the expected case where the boxcar coherence is lower than the sibling coherence. We suggest this may be due to the intensity of the damage caused by this event, both in terms of the landslides, which had a much higher density than for Nepal or Lombok, and possibly also in terms of vegetation damage caused by Typhoon Jebi. Since the Bx-S method relies on siblings of a landslide pixel lying outside the landslide, it may not work well in the case where many landslides are close together. This is similar to the case found by Burrows et al. (2019), where the Bx-S method was unable to identify large landslides if the sibling search window was not sufficiently large.

## 5.3 Future work

We have demonstrated that the classification ability of a method can vary significantly based on the data used, the nature of the triggering event, and the resolution of the analysis. For example, the Bx-S method performed significantly better in the case of Nepal than Hokkaido, and CECL was up to 32% more successful when using ALOS-2 compared to Sentinel-1 data with the same event and look direction. Thus, it is clear that any future SAR-based classification method must be widely tested on multiple events, and it cannot be assumed that a method performing well with one SAR dataset will be equally successful using data from a different sensor.

While we tested ascending and descending tracks separately, a more complete picture would be obtained through combining these two tracks, as hillslopes that are unfavourably oriented to the satellite in one track are likely to be more favourably oriented to the other track. A complete map would be easier to interpret for emergency responders than two maps, each of which is missing landslides on unfavourably oriented slopes. This is particularly important since landslides are most likely on steep slopes where these orientation effects are likely to be most severe.

Although we selected earthquakes to use as case studies, SAR methods could be equally useful in the case of rainfall-triggered landslides, where cloud cover may also cause delays to mapping using optical satellite imagery. We have shown that SAR methods performed well in Hokkaido, where the co-event coherence maps spanned both an earthquake and a typhoon. This demonstrates that the alterations in SAR coherence and amplitude that can result from changes in soil moisture content are not prohibitive to the application of SAR methods to rainfall triggered landslides, although some false positives may have been caused by wind damage.

The Lombok case studies demonstrate the importance of knowing the timing of the landslides when applying current methods. When moving from Lombok-1, in which all landslides were assumed to have been triggered by a single earthquake, to Lombok-2, which contained landslides triggered by a series of earthquakes over a two-week period, it was necessary to in-

505 crease the timespan of the Sentinel-1 co-event interferograms from 6 days to 24 days on track S032d and 18 days on track S156a, which significantly reduced the AUC of the classification methods. Therefore, if we wish to apply radar methods more widely to landslide triggering events that are dispersed through time, such as long rainfall events (e.g., those associated with the monsoon) or earthquake sequences, more work will be required on the characterisation of the signals of landslides in radar imagery through time.

Finally, here we assessed classifier performance using ROC analysis, which does not require a threshold to be applied to the classification surface. However, if SAR methods are to be applied to future events for emergency response, it will be necessary to set a threshold between 'landslide' and 'non-landslide'. In this study, the time between image acquisitions varied significantly between events, making it unlikely that a threshold could be selected that would work well for both Sentinel-1 and ALOS-2 across all events. However, this may be possible in the future when more events have been studied and SAR data with 515 more regular acquisitions are available. Further work is therefore needed to establish such thresholds, which will be determined according to the requirements of emergency responders and their relative tolerance for false positives and false negatives.

## 6 Conclusions

We have demonstrated that it is possible to generate landslide density information from SAR coherence within a typical two-week emergency response time-frame and at a useful spatial resolution. The best performing method is dependent on the 520 wavelength of the available imagery and the number of post-event images available during the emergency response. The CECL method is the best performing method when only one post-event L-band image is available (ROC AUC = 0.7-0.89), and this method could be applied within two weeks for three out of our four case study events. However, Sentinel-1 data were available earlier than ALOS-2 for all of the events studied here, with the first image available within 4 days of each earthquake. Using the first post-event Sentinel-1 image acquired, the best method is the Bx-S method with AUC between 0.58-0.74. Methods that 525 use a second post-event image improve overall accuracy by an average of 10 % and are more consistently reliable across event and sensor type. These approaches could be valuable for landslide mapping when latency is less important, but the additional waiting time for the second post-event image is a disadvantage in an emergency response situation.

*Author contributions.* KB carried out data curation, investigation and formal analysis of the data and visualisation and writing of the original draft. Conceptualization, administration, funding acquisition and supervision of the project were carried out by RJW, DM and ALD. All 530 authors were involved in reviewing and editing the manuscript and in the methodology.

*Competing interests.* The authors declare no conflict of interest.

*Acknowledgements.* Sentinel-1 interferograms and coherence maps are a derived work of Copernicus data, subject to ESA use and distribution conditions. The authors are grateful to JAXA for providing ALOS-2 data sets under the research contract of RA-6 (PI No. 3228). Some figures were made using the public domain Generic Mapping Tools (Wessel and Smith, 1998). The work presented here was carried out as part of a PhD project funded by the Institute of Hazard, Risk and Resilience, Durham University, through an Action on Natural Disasters scholarship. This work was partly supported by the UK Natural Environmental Research Council (NERC) through the Centre for the Observation and Modelling of Earthquakes, Volcanoes and Tectonics (COMET) and grants NE/J01995X/1 and NE/N012216/1. We thank Karsten Spaans for assistance in using the RapidSAR software.

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

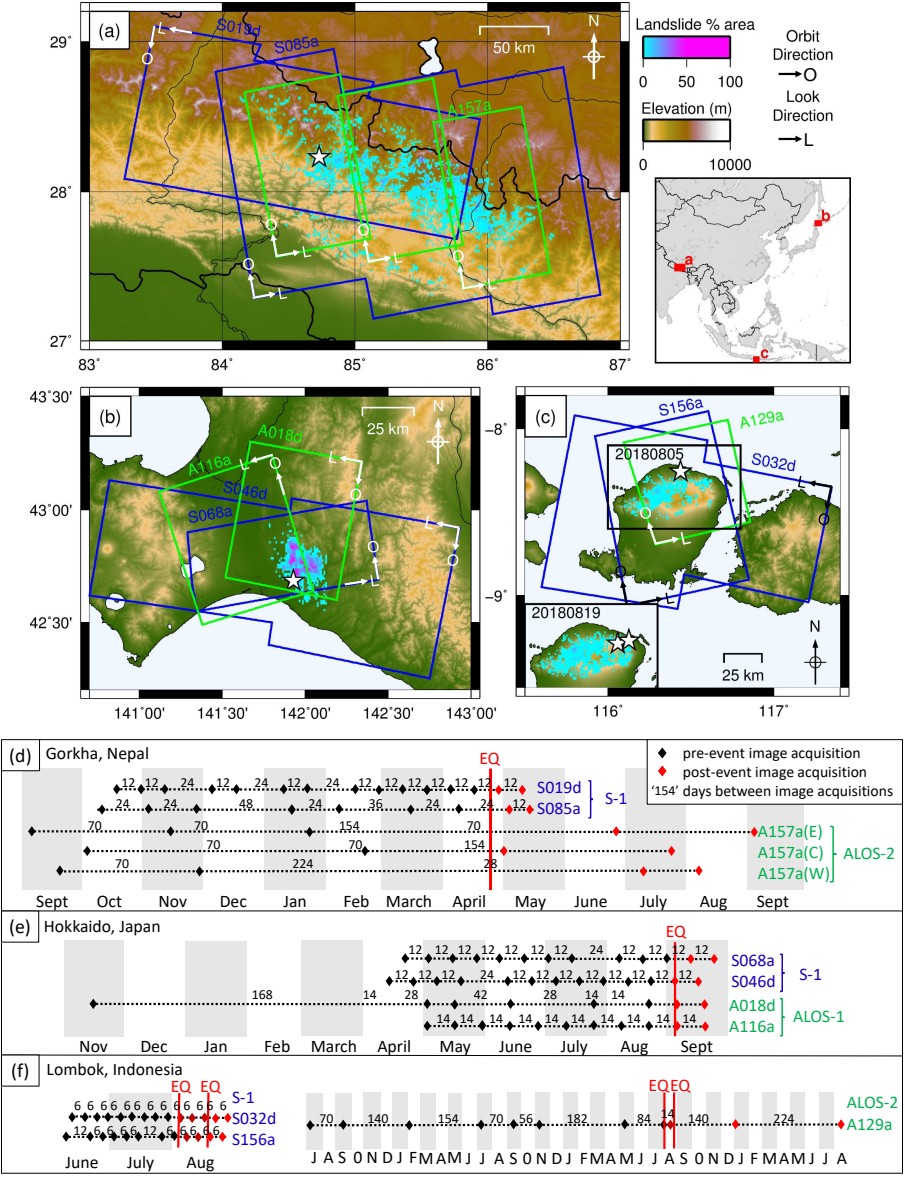

**Figure 1.** (a, b, c) Landslides triggered by the 2015 Gorkha, Nepal earthquake, the 2018 Hokkaido, Japan earthquake, and the 2018 Lombok, Indonesia earthquakes, respectively, plotted as the areal density of landsliding based on 1 km² cells (Roback et al., 2018; Zhang et al., 2019; Ferrario, 2019). ALOS-2 SAR acquisitions are shown in green and Sentinel-1 in blue. In (c), a second earthquake, referred to as "Lombok-2" is inset. (d, e, f) Acquisition dates and track numbers of SAR imagery used in this study. Earthquakes are shown as red vertical lines, black symbols show pre-event image acquisition dates, and red symbols show post-event dates.

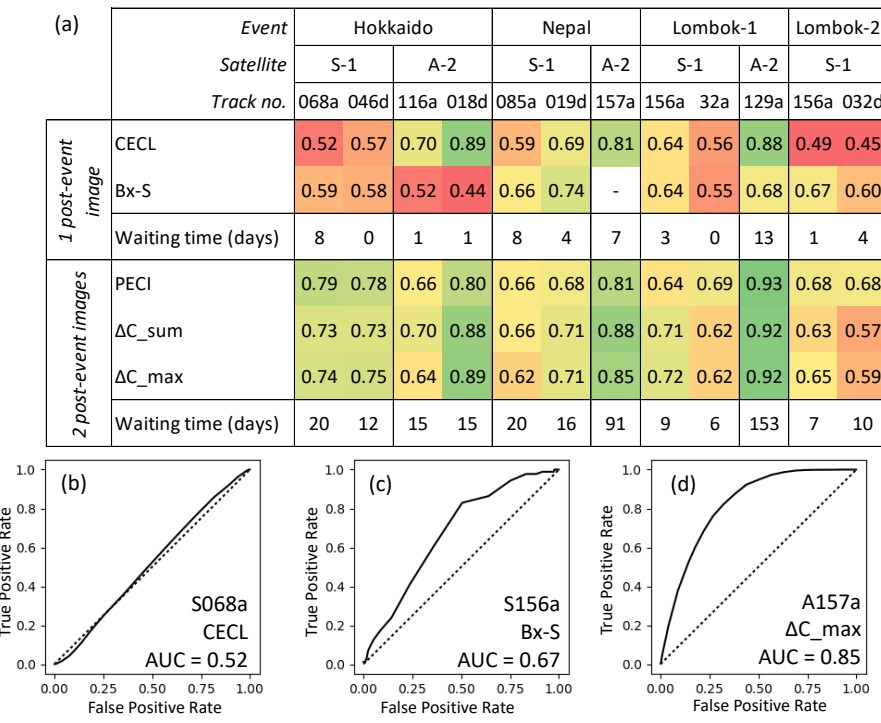

**Figure 2.** (a) AUC values for each classifier described in Section 2 at a resolution of 200 m × 220 m. For Hokkaido, we use the inventory of Zhang et al. (2019). Insufficient pre-event data were available on A157a to calculate the Bx-S classification surface. Colours range from red (worst performing AUC < 0.55) to green (best performing, AUC >0.80). ( b, c, d) Examples of ROC curves for (b) the CECL method on track S156a, Hokkaido; (c) the Bx-S method on track S156a, Lombok-2; (d) ΔC_max on track A157a, Nepal.

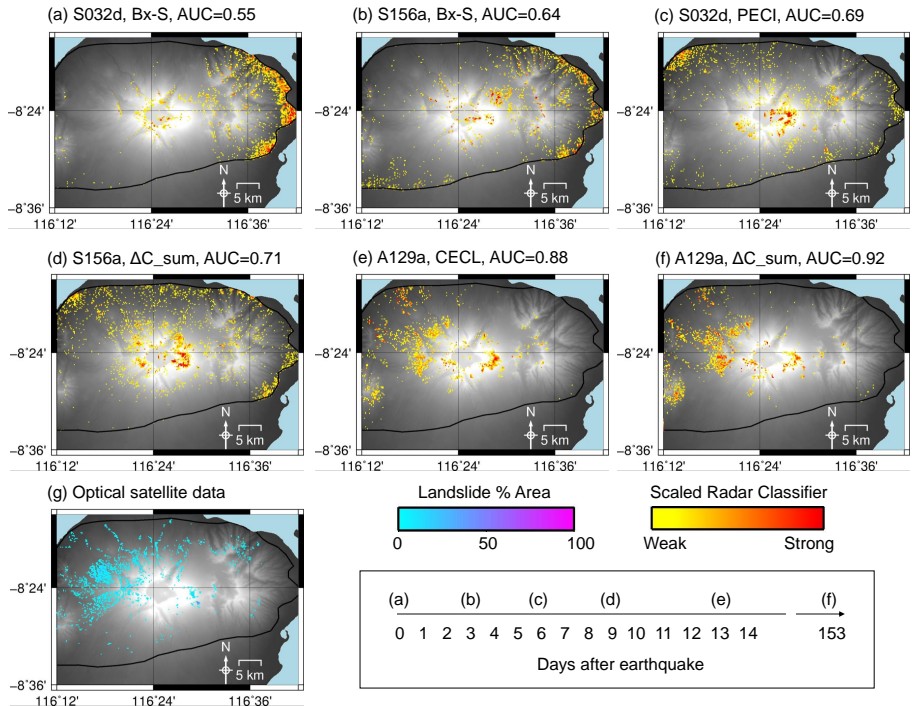

**Figure 3.** (a-f) Time series of classification surfaces in the order that SAR data were acquired on tracks S032d, S156a and A129a following the 5 August 2018 Lombok earthquake, using the methods we recommend in Section 4.1.(g) Observed Landslide areal density for 200 m x 220 m pixels (calculated from Ferrario, 2019).

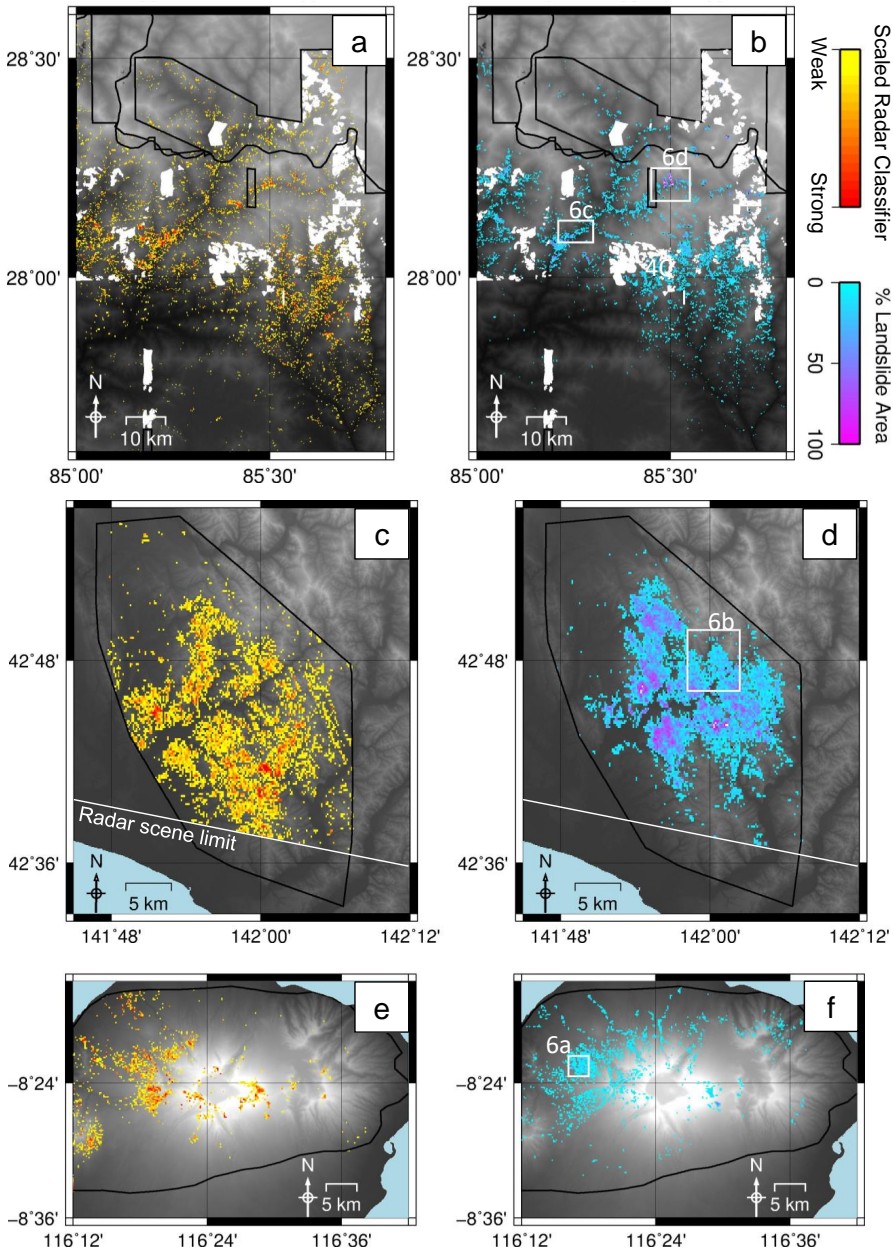

**Figure 4.** SAR-based landslide classification surfaces for the Gorkha (a), Hokkaido (c) and Lombok-1 (e) earthquakes calculated with ALOS-2 radar data using the $\Delta C\_sum$ method at a 200 m × 220 m resolution. (b, d, f) Observed landslide density calculated as the percentage of each 200 m × 220 m cell covered by landsliding for each event according to the inventories of Ferrario (2019), Roback et al. (2018), Zhang et al. (2019) respectively. Cells where landslide density was 0 were masked. For the $\Delta C\_sum$ surface, a threshold value of the classifier was selected such that the number of cells plotted in panels (a, c, e) for each event was the same as the number plotted in panels (b, d, f)

| | | Hokkaido | | | | Nepal | | | Lombok | | | Lombok 2 | | *Event* |
|---|---|---|---|---|---|---|---|---|---|---|---|---|---|---|
| | | S-1 | | A-2 | | S-1 | | A-2 | S-1 | | A-2 | S-1 | | *Satellite* |
| | | 68A | 46D | 116A | 18D | 85A | 19D | 157A | 156A | 32D | 129A | 156A | 32D | *Track number* |
| **1 post-event image** | CECL | 0.51 | 0.51 | 0.62 | 0.72 | 0.52 | 0.58 | 0.68 | 0.53 | 0.50 | 0.76 | 0.50 | 0.49 | |
| | Bx-S | 0.52 | 0.52 | 0.52 | 0.51 | 0.56 | 0.59 | - | 0.54 | 0.52 | 0.69 | 0.53 | 0.52 | |
| | Waiting time | 8 | 0 | 1 | 1 | 8 | 4 | 7 | 3 | 0 | 13 | 1 | 4 | |
| **2 post-event images** | PECI | 0.60 | 0.61 | 0.60 | 0.65 | 0.54 | 0.55 | 0.67 | 0.53 | 0.54 | 0.76 | 0.53 | 0.55 | |
| | ΔC_sum | 0.57 | 0.58 | 0.63 | 0.71 | 0.54 | 0.57 | 0.72 | 0.54 | 0.52 | 0.79 | 0.52 | 0.52 | |
| | ΔC_max | 0.58 | 0.59 | 0.60 | 0.72 | 0.53 | 0.57 | 0.70 | 0.54 | 0.53 | 0.80 | 0.52 | 0.54 | |
| | Waiting time | 20 | 12 | 15 | 15 | 20 | 16 | 91 | 9 | 6 | 153 | 7 | 10 | |

**Figure 5.** AUC values for each classifier described in Section 2 at a resolution of 20 m × 22 m. For Hokkaido, we use the inventory of Zhang et al. (2019). Insuffcent pre-event data were available on A157a to calculate the Bx-S classification surface. Colours range from red (worst performing AUC < 0.55) to green (best performing, AUC >0.80).

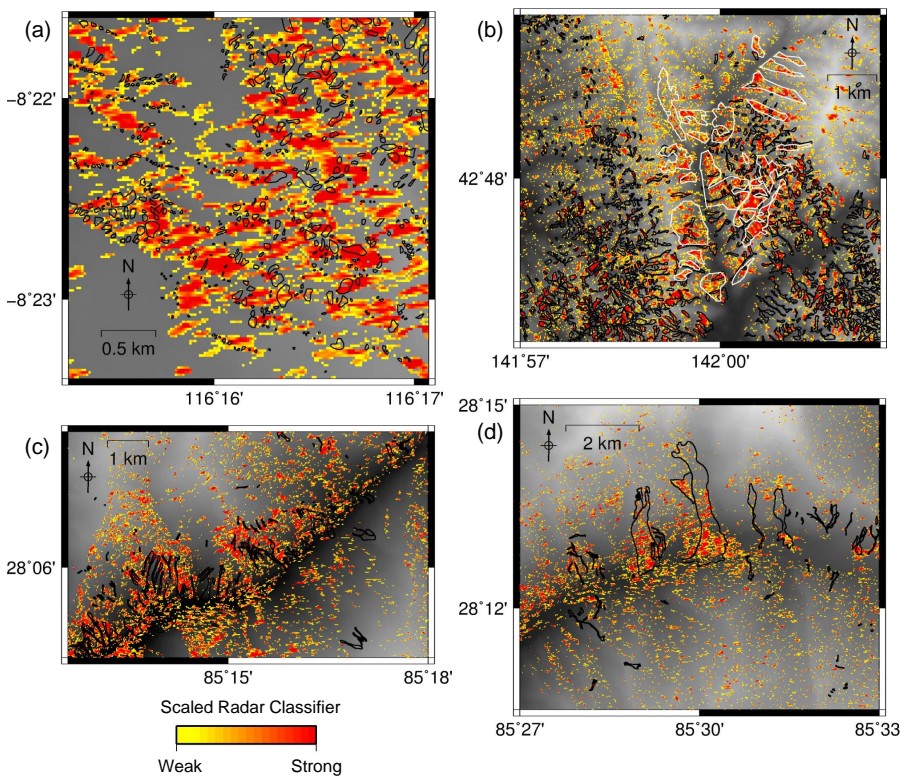

**Figure 6.** ΔC_sum classification surface calculated using ALOS-2 data displayed at 20 m x 22 m resolution with mapped landslide polygons in black. The 10% of pixels most likely to be landslide pixels are coloured from yellow (least likely) to red (most likely) (a) Lombok: track A129a for the 5 August 2018 event (Ferrario, 2019). (b) Hokkaido: track A018d, landslide polygons from Zhang et al. (2019). White polygons show the locations of forested areas mapped using Sentinel-2 data. (c,d) Nepal: track A157a(C), landslide polygons from Roback et al. (2018).

**Table 1.** The resolution and pixel spacing of the data at different stages throughout the processing for Sentinel-1, ALOS-2 on tracks A018d and A116a (Hokkaido), track A157a (Nepal) and track A129a (Lombok). Resolutions are given in Range $\times$ Azimuth coordinate system

| | Resolution (m) | Radar pixel spacing (m) | Multilooked radar pixel spacing (m) | Boxcar window size (m) | Sibling search window size (m) |
|---|---|---|---|---|---|
| All Sentinel-1 | 20 × 22 | 2.3 × 14.0 | 11.6 × 14.0 | 34.8 × 42.0 | 940 × 1134 |
| ALOS-2 018d | 3 × 3 | 1.4 × 2.1 | 7.2 × 10.6 | 21.6 × 31.8 | 583 × 859 |
| ALOS-2 116a | 3 × 3 | 1.4 × 1.9 | 7.2 × 9.3 | 21.6 × 27.9 | 583 × 753 |
| ALOS-2 157a | 10 × 10 | 4.3 × 3.8 | 21.5 × 19 | 64.5 × 57.0 | 1742 × 1539 |
| ALOS-2 129a | 10 × 10 | 4.3 × 3.3 | 21.5 × 16.3 | 64.5 × 48.9 | 1742 × 1320 |