# Peer review of "A Systematic Exploration of Satellite Radar Coherence Methods for Rapid Landslide Detection"

_Natural Hazards and Earth System Sciences, 2020_

## Referee Comment (RC1) · Anonymous Referee #1 · 13 Jul 2020

The paper is well written, and the results clearly presented. The topic is of interest for the readers since the research on radar-based post event landslide mapping based on coherence/amplitude is still ongoing. My minor comments are listed below. Abstract L5, "triggering events". You can specify the type of event L6 "ARIA", specify the acronym L10 "useful landslide density", I would say landslide mapping Introduction L22 "could have been much longer had the earthquake occurred during Nepal's monsoon season", check this L30 "reliability of global empirical rainfall-triggered landslide susceptibility maps", the scale itself is a limitation L31 "style of landsliding,", I would say landslide type L41 there are thousands of published paper in this field. I think the reference list can be extended, see some recent examples: _Dai, K., Li, Z., Tomás, R., Liu, G., Yu, B., Wang, X., ... & Stockamp, J. (2016). Monitoring activity at

the Daguangbao mega-landslide (China) using Sentinel-1 TOPS time series interferometry. Remote Sensing of Environment, 186, 501-513. _Solari, L., Del Soldato, M., Raspini, F., Barra, A., Bianchini, S., Confuorto, P., ... & Crosetto, M. (2020). Review of Satellite Interferometry for Landslide Detection in Italy. Remote Sensing, 12(8), 1351. _Aslan, G., Foumelis, M., Raucoules, D., De Michele, M., Bernardie, S., & Cakir, Z. (2020). Landslide Mapping and Monitoring Using Persistent Scatterer Interferometry (PSI) Technique in the French Alps. Remote Sensing, 12(8), 1305. _Reyes-Carmona, C., Barra, A., Galve, J. P., Monserrat, O., Pérez-Peña, J. V., Mateos, R. M., ... & Azañón, J. M. (2020). Sentinel-1 DInSAR for Monitoring Active Landslides in Critical Infrastructures: The Case of the Rules Reservoir (Southern Spain). Remote Sensing, 12(5), 809. _Hu, X., Bürgmann, R., Lu, Z., Handwerger, A. L., Wang, T., & Miao, R. (2019). Mobility, thickness, and hydraulic diffusivity of the slow‐moving Monroe landslide in California revealed by L‐band satellite radar interferometry. Journal of Geophysical Research: Solid Earth, 124(7), 7504-7518. Satellite radar coherence for change detection L68 "method"–> technique L77 any reference for the boxcar method? Case studies L131 "landslides were triggered by the earthquake", the occurrence of a typhoon the day before is certainly a trigger as well. L140 and followings, what about the type of landslides that have been triggered? Fig.1 is the inset in (c) Lombok 2? You should specify this in the caption Landslide detection methods L185 check the double "by a landslide" Recommendations on data and methods L379 and one day with images from ROSE-L Building damage L398 do you have an estimate of the number/percentage of false positives due to the buildings? The same comment for the number of false positives due to wind or snow. Rivers You could mask the rivers. Since the single landslide is not the target of your approach, the detection of a single landslide dam should not be possible and the river-mask duable.

---

## Referee Comment (RC2) · Anonymous Referee #2 · 3 Aug 2020

General comments:

The authors present a quantitative comparison of multiple different methods of mapping landslides using the coherence of synthetic aperture radar images. The quantitative comparison between different methods, with tests in different regions and examination of the local features that can affect results, is particularly valuable given the ongoing work on using SAR for disaster response. The work is well written and figures are of a good standard.

There are some places where I think the assumptions underlying certain choices, both for the landslide mapping methods and the quantification of the results could be better explained; I have provide further comments on these below. There are also numerous places where I think small tweaks to the wording and slightly more explanations would

be helpful to the reader.

Technical comments:

Quantification of different methods:

I would like to see further discussion of the use of the ROC curve to evaluate these classification methods. My understanding is that ROC curves are generally best deployed when the dataset is balanced between positive and negative examples (i.e. roughly equal numbers of landslide and non-landslide pixels), however I wonder if that is the case here, or if there are many more non-landslide pixels than landslide pixels in general?

If it's the case that the dataset is imbalanced, with more non-landslide pixels, then the ROC curve might not be the best way to assess how well your method is performing. For example, if I take a balanced dataset then add a large number of negative examples, with their classification scores drawn from the same distribution at the existing negative examples, the false positive rate (FPR=FP/(FP+TN)) and true positive rate (TPR=TP/(TP+FN)) remain the same for a given threshold, and the ROC curve doesn't change. However I now have many more false positives at a given threshold, meaning the precision of my classifier (=TP/(TP+FP)) will decrease (i.e. a smaller fraction of my positive classifications will be true positives). In some circumstances it could be the case that only a small fraction of my samples that are classified as positive are actually true positives, even as my TPR and FPR appear good.

Additionally, as I expand the region spanned by the SAR data, it's possible that I include more and more pixels that will have lower noise levels, e.g. as they're further away from the earthquake and so have less of the building damage, surface rupture, liquefaction etc. that can lead to false positives for landslide classification. This would lead to having a larger number of true negatives, and so an improved false positive rate, thus an improved ROC curve, but would have minimal effect on the precision, which doesn't consider true negatives.

In addition to the currently presentation, you should consider presenting a precision-recall curve, or mention why the the ROC curve is preferred over the precision-recall curve. It would also be good to have you mention the fraction of each region that is covered with landslide and non-landslide pixels.

Histogram matching:

The use of histogram matching between the pre- and co-seismic coherence images could do with some further explanation so the reader understands the motivation and assumptions. My understanding is that this adjustment assumes that only a small number of pixels are anomalous (i.e. contain landslides)—otherwise you would end up removing the signal you were looking for. Furthermore, I think there is an assumption that the coherence of the pixels in the image that's adjusted have all been affected in the same way. For example, if only the southern part of the image had been covered in snow between the final pre-seismic and first post-seismic SAR images, then adjusting the entire coseismic coherence image based on a simple matching of histograms would not be the correct approach, however if the second coherence image just had a longer temporal baseline then the extra temporal decorrelation might be removable by histogram matching.

Optimum thresholds:

It might be helpful for the discussion to go into more detail on what the optimum threshold for flagging landslides would be for each method, as this is what would be required before use by a first responder. Currently the presentation of your results is threshold free (i.e. in terms of the ROC curve), apart from when you make plots. The discussion around line 300-305 explains that you choose a threshold for plotting in order to be consistent with the number of points plotted in the landslide area plot in Fig 4, but it's not clear if this would be a useful threshold for response. The comparison you chose to present in figure 4 was also slightly confusing to me; in line 236 you say that you're only classifying pixels as 'landslide' pixels if they have >25% area of landslides, however for

the plot in figure 4 you plot pixels that have >1% area of landslides, and then choose the classification threshold based on the number of pixels that have >1% landslide area. Is there a reason for the different threshold of landslide area? Would it not be better to choose the classification threshold from some desired trade off between true positive and false positive rates on the ROC curve (or a trade off between precision and recall)?

'ARIA' method naming:

The use of the descriptor 'ARIA' method isn't standard to my knowledge, and may be misleading. JPL's ARIA Project does employ this method, although it is also working on many alternative approaches. Furthermore, the method, or similar variants of it, has been used by many groups (e.g. Fieldling et al (2005)) and is not referred to as the 'ARIA' method in these publications. One term I have seen applied is 'Coherence Change Detection' (e.g. Washaya et al. 2018), however this may not be the most useful, as all of your methods are change detection using coherence. Perhaps 'coherence loss' would be a better term?

Citations for the paragraph above:

- Fielding, E. J. et al. (2005) 'Surface ruptures and building damage of the 2003 Bam, Iran, earthquake mapped by satellite synthetic aperture radar interferometric correlation', Journal of Geophysical Research: Solid Earth, 110(3), pp. 1–15. doi: 10.1029/2004JB003299. - Washaya, P., Balz, T. and Mohamadi, B. (2018) 'Coherence Change-Detection with Sentinel-1 for Natural and Anthropogenic Disaster Monitoring in Urban Areas', Remote Sensing, 10(7), p. 1026. doi: 10.3390/rs10071026.

Line by Line comments:

Line 73 - ïż£'The signal-to-noise ratio of each pixel in an InSAR image is described by its coherence'. This statement feels incomplete to me—in your work the 'signal' is the decorrelation of the pixel, and in circumstances where the phase is the signal (e.g.

deformation time series), the decorrelation is only one contributor to the noise (e.g. can also have atmospheric and ionospheric noise which doesn't affect the coherence).

Line 77 - I think some mention of the assumptions of the box-car and sibling methods would be useful. My understanding is that the box-car method assumes that the surrounding pixels are statistically similar, and the sibling method identifies pixels that are similar through time (based on the amplitude) and assumes that these remain similar for subsequent acquisitions. I note that the sibling method is briefly explained on line 212, but more information in the introduction I feel would be helpful to the reader.

Line 87 - perhaps include a definition of perpendicular baseline, and mention that orbital controls on modern satellites have rendered this a much smaller issue?

Line 100 - add citation on the effect of vegetation on coherence at different wavelengths?

Line 106 'as it is impossible to combine data from both tracks' - it would be helpful to clarify what is meant by 'combine' (i.e. you can't calculate the coherence between different tracks)

Line 119 - what is meant by 'high resolution' data for ALOS-2? Might be helpful to mention the acquisition mode for clarity?

Line 165 - need to clarify that the user must choose a coherence loss threshold for flagging damage, the current sentence structure implies that all pixels that have any coherence decrease at all are flagged as damaged

Line 179 - I think it's worth reminding the reader here that the sibling pixels have been specifically identified to be behaving similarly before the earthquake by looking at the amplitude time series, and it's on that basis that they're expected to behave similarly

Line 185 - post-event coherence - make it clear that this coherence is calculated using the boxcar method (which I assume it is?)

Line 233 - 'If over 95% of an aggregate pixel is made up of masked pixels, the aggregate pixel was masked' - how do you choose this number? Are you results sensitive to this choice? What fraction of pixels end up being masked out?

Line 234 - ïż£'In this study, we did not attempt to map SAR classification surface values directly to landslide areal density values, as this has not been attempted in previous studies and may not be possible' - would be good to get more info about why this is/isn't possible, and a citation for the previous work. Is this comment based on the work of Burrows et al. (2019)?

Line 236 - how did you select the 25% threshold? How much does this choice affect your results? How large is this area, and what fraction of the total landslide area is missed by virtue of the fact that it falls below this threshold? Would be helpful to have these questions discussed in the text, particularly if this choice has a large affect on AUC values.

Line 277 - 'ïż£the 6-day Sentinel-1 acquisition window' - this is a little unclear, to my mind 'window' implies that data was acquired at some point in that time range. Rather than 'window' could say 'acquisition frequency'?

Line 287 - for clarity it may be worth reminding the reader that the L-band image is from the ALOS 2 satellite

Line 324 - a brief discussion of what is meant by 'higher quality siblings' would be helpful (I imagine that they're more statistically similar?)

Line 425 - You mention rivers giving false positives for the Bx-S method. Would it be possible to identify rivers by applying the Bx-S method to a pre-seismic coherence image (using the same pixel siblings) and then use that as a mask on your co-seismic landslide image? You could propose this if so.

Line 469 - ïż£'The ARIA method is the best performing method when only one L-band image is available' - does this mean 'one post-seismic L-band SAR image'? I think

this could be clarified, as I could read this to mean that only on L-band SAR image is available in total. Figure 6 - how is the threshold for these plots chosen? Could you clarify in the caption?

Technical Corrections:

For words in quotation marks apostrophes are being used throughout, rather than opening and closing quotation marks. Please adjust.

Line 185 - double use of 'by a landslide' in this sentence.

Line 235 and 443 - add a comma after 'Thus' (consistent with 'Thus' on line 10)

Table 1 - 'x' has inconsistent formatting or is missing in some cases

---

## Author Comment (AC1) · 23 Sep 2020

We thank the reviewer for their helpful suggestions and have responded to each of their comments below.

Abstract L5, "triggering events". You can specify the type of event

We will change this to "triggering earthquakes"

L6 "ARIA", specify the acronym

Thank you for identifying this. We are renaming this method based on comments from the other reviewer so this will be changed to "co-event coherence loss method"

L10 "useful landslide density", I would say landslide mapping

We used landslide density so that readers do not expect individual mapped landslides, which is the most common output from mapping landslides using optical satellite imagery.

Introduction

L22 "could have been much longer had the earthquake occurred during Nepal's monsoon season", check this

This statement is supported by Robinson et al. 2019, which is referenced earlier in the same sentence.

L30 "reliability of global empirical rainfall-triggered landslide susceptibility maps", the scale itself is a limitation

The spatial scale of the empirical models may be a disadvantage for some applications, but immediately after the earthquake, low resolution information over a large spatial scale is useful for giving an overview of severely impacted areas to emergency response coordinators (Discussed in Williams et al. NHESS. 2018). Although our analysis is carried out at a slightly higher resolution than these empirical models (200 x 220 m pixels as opposed to 1 km2, which is used by Nowicki-Jessee), the improvement in resolution in this study is fairly limited. We have added a sentence on the resolution of the empirical models "The outputs of these models have a comparatively low spatial resolution (around 1 km2 in the case of Nowicki Jessee et al. 2018), but can be used to provide an overview of the most severely impacted areas."

L31 "style of landsliding,", I would say landslide type

We have changed this to "type of landslides".

L41 there are thousands of published paper in this field. I think the reference list can be extended, see some recent examples: _Dai, K., Li, Z., Tomás, R., Liu, G., Yu, B., Wang, X., ... & Stockamp, J. (2016). Monitoring activity at the Daguangbao mega-landslide (China) using Sentinel-1 TOPS time series interferometry. Remote Sensing of Environment, 186, 501-513. _Solari, L., Del Soldato, M., Raspini, F., Barra, A., Bianchini, S., Confuorto, P., ... & Crosetto, M. (2020). Review of Satellite Interferometry for Landslide Detection in Italy. Remote Sensing, 12(8), 1351. _Aslan, G., Foumelis, M., Raucoules, D., De Michele, M., Bernardie, S., & Cakir, Z. (2020). Landslide Mapping and Monitoring Using Persistent Scatterer Interferometry (PSI) Technique in the French Alps. Remote Sensing, 12(8), 1305. _Reyes-Carmona, C., Barra, A., Galve, J. P., Monserrat, O., Pérez-Peña, J. V., Mateos, R. M., ... & Azañón, J. M. (2020). Sentinel-1 DInSAR for Monitoring Active Landslides in Critical Infrastructures: The Case of the Rules Reservoir (Southern Spain). Remote Sensing, 12(5), 809. _Hu, X., Bürgmann, R., Lu, Z., Handwerger, A. L., Wang, T., & Miao, R. (2019). Mobility, thickness, and hydraulic diffusivity of the slowâĚŸARĚĞmoving Monroe landslide in California revealed by LâĚŸARĚĞ band satellite radar interferometry. Journal of Geophysical Research: Solid Earth, 124(7), 7504-7518.

Thank you, these have now been added to the list of references at L41.

Satellite radar coherence for change detection

L68 "method"–> technique

Thank you. We have changed this in the text.

L77 any reference for the boxcar method?

This is the most common method of calculating coherence, We have altered the text to make this clearer and added some examples of texts that use this method.

Proposed new text: "The ensemble is chosen so that the pixels used in the calculation are expected to be similar. In a "boxcar" method, it is assumed that pixels immediately adjacent to and centred on the target pixel are similar to it (e.g. Hansen, 2001; Yun et al. 2015)."

Case studies

L131 "landslides were triggered by the earthquake", the occurrence of a typhoon the day before is certainly a trigger as well.

We agree that the typhoon the day before the earthquake will have caused widespread pore-pressure increase across the study area and that this may have played an important role in the initiation of some of the observed landslides. However, the landslides have been widely attributed to the earthquake as the primary trigger, with debate over the role of rainfall being focussed on the extent to which it pre-conditioned the slopes for failure (e.g. Osanai et al., 2019; Wang et al., 2019; Yamagishi and Yamazaki, 2018; Zhang et al., 2019). We are interested in this event because of the close sequencing of intense rain and shaking rather than as a test of our ability to distinguish the different triggers. We propose to remove reference to the earthquake as the trigger since this is unnecessary and replace that sentence with the following text:

Proposed new text : "...with the advantage that because the typhoon and earthquake occurred one day apart, and aerial imagery of the triggered landslides was acquired immediately afterwards, we know more precisely when the landslides occurred (Yamagishi and Yamazaki, 2018)."

References

Osanai, N., Yamada, T., Hayashi, S.I., Kastura, S.Y., Furuichi, T., Yanai, S., Murakami, Y., Miyazaki, T., Tanioka, Y., Takiguchi, S. and Miyazaki, M., 2019. Characteristics of landslides caused by the 2018 Hokkaido Eastern Iburi Earthquake. Landslides, 16(8), pp.1517-1528.

Wang, F., Fan, X., Yunus, A.P., Subramanian, S.S., Alonso-Rodriguez, A., Dai, L., Xu, Q. and Huang, R., 2019. Coseismic landslides triggered by the 2018 Hokkaido, Japan (M w 6.6), earthquake: spatial distribution, controlling factors, and possible failure mechanism. Landslides, 16(8), pp.1551-1566.

Yamagishi, H. and Yamazaki, F., 2018. Landslides by the 2018 Hokkaido Iburi-Tobu

Earthquake on September 6. Landslides, 15(12), pp.2521-2524. Zhang, S., Li, R., Wang, F. and Iio, A., 2019. Characteristics of landslides triggered by the 2018 Hokkaido Eastern Iburi earthquake, Northern Japan. Landslides, 16(9), pp.1691-1708.

L140 and followings, what about the type of landslides that have been triggered?

We will add the following information at line 119

"Finally, the type of landslides triggered by our four case study earthquakes were typical of landslides triggered by earthquakes (Keefer, 1984). The majority of ground failures in the four earthquakes were slides. In Nepal, ground failures were primarily a mixture of slides and falls, with the exception of a large debris avalanche in the Langtang Valley. For all four earthquakes, failure surfaces were at shallow depths in most cases with a small number of exceptions (Collins and Jibson, 2015; Ferrario, 2019; Yamagishi and Yamazaki, 2018)."

Fig.1 is the inset in (c) Lombok 2? You should specify this in the caption

Thank you for pointing this out, we have added this information to the caption.

Landslide detection methods

L185 check the double "by a landslide"

Thank you, this has been fixed

Recommendations on data and methods

L379 and one day with images from ROSE-L

Thank you, I was not aware of this satellite and have added a reference to this misson.

Building damage

L398 do you have an estimate of the number/percentage of false positives due to the buildings? The same comment for the number of false positives due to wind or snow.
It is difficult to quantify the number of false positives resulting from a specific source as we do not threshold the classifiers in our analysis. However, we have tried to assess the impact of buildings by repeating the ROC analysis with urban areas masked based on the ESA CCI landcover map. We found very little change in ROC AUC (<0.02), as these land cover types make up a relatively small part of the study area.

Quantifying false positives in snow-covered areas is difficult, since these are also likely to be underrepresented in the landslide inventory generated from optical satellite imagery (Roback et al. 2018). Furthermore, the dependence of snow cover on season, elevation and climate means that any measure we obtain of false positives due to snow is unlikely to be applicable in future events.

Quantification of false positives due to wind damage is also difficult as there are no reliable validation data available for this. From visual inspection, it appeared that the signal was strongest for areas mapped as "evergreen needleleaf forest" but masking these areas did not noticeably improve ROC AUC (<0.02).

It is difficult to predict how these false positives might affect the applicability of SAR data to future events, as this will depend on the extent of urban development (in the case of building damage), the climate, elevation and season (in the case of snow) and on the occurrence of storms (in the case of wind damage). Therefore, while we note that these are potential causes of error, we do not think it is useful to try to quantify these further.

Rivers

You could mask the rivers. Since the single landslide is not the target of your approach, the detection of a single landslide dam should not be possible and the river-mask duable.

We agree this would be possible, but although we cannot map individual landslide dams, points where rivers pass through areas of high landslide density can be used to

identify locations where a landslide dam is more probable (e.g. Robinson et al. BSSA, 2018) and therefore which are worthy of further investigation. Therefore, although we acknowledge that a landslide signal from SAR data is less reliable in a river than elsewhere, masking them would lead to a more confusing product where landslides are not allowed to intersect with rivers. Instead we suggest that rivers should be drawn over the coherence products. In response to the comments in this review and the following comment made by the other reviewer "Line 425 - You mention rivers giving false positives for the Bx-S method. Would it be possible to identify rivers by applying the Bx-S method to a pre-seismic coherence image (using the same pixel siblings) and then use that as a mask on your co-seismic landslide image? You could propose this if so.", we will add the following sentences at line 429:

"A variety of methods could be used to identify and remove rivers from our analysis (including using a pre-event Bx-S surface). However, since areas where landslides and rivers intersect are particularly hazardous due to the potential for landslide dams and associated flash flooding, we did not mask rivers in this study. We suggest that any product based on SAR coherence supplied to emergency response coordinators should have rivers overlaid. This would both mask false positives due to rivers and allow identification of locations where rivers pass through areas of intense landsliding."

Please also note the supplement to this comment:
https://nhess.copernicus.org/preprints/nhess-2020-168/nhess-2020-168-AC1-supplement.pdf
* * *

---

## Author Comment (AC2) · 23 Sep 2020

We thank the reviewer for their helpful comments. Our response is attached to this review as a supplement to preserve the formatting.

Please also note the supplement to this comment:
https://nhess.copernicus.org/preprints/nhess-2020-168/nhess-2020-168-AC2-supplement.pdf

| | Event | Hokkaido | | | | Nepal | | | Lombok | | | Lombok 2 | |
|---|---|---|---|---|---|---|---|---|---|---|---|---|---|
| | Satellite | S-1 | | A-2 | | S-1 | | A-2 | S-1 | | A-2 | S-1 | |
| | Track number | 068a | 046d | 116a | 018d | 085a | 019d | 157a | 156a | 032d | 129a | 156a | 032d |
| 1 post-event image | ARIA | 0.103 | 0.099 | 0.179 | 0.296 | 0.005 | 0.002 | 0.045 | 0.002 | 0.001 | 0.009 | 0.002 | 0.004 |
| | Bx-S | 0.120 | 0.116 | 0.103 | 0.084 | 0.008 | 0.009 | - | 0.002 | 0.001 | 0.009 | 0.010 | 0.004 |
| | Waiting time (days) | 8 | 0 | 1 | 1 | 8 | 4 | 7 | 3 | 0 | 13 | 1 | 4 |
| 2 post-event images | PECI | 0.271 | 0.273 | 0.143 | 0.195 | 0.009 | 0.005 | 0.022 | 0.015 | 0.003 | 0.018 | 0.005 | 0.005 |
| | ΔC_sum | 0.180 | 0.191 | 0.179 | 0.257 | 0.006 | 0.002 | 0.057 | 0.003 | 0.002 | 0.016 | 0.004 | 0.005 |
| | ΔC_max | 0.206 | 0.211 | 0.143 | 0.295 | 0.007 | 0.002 | 0.051 | 0.014 | 0.002 | 0.014 | 0.004 | 0.005 |
| | Waiting time (days) | 20 | 12 | 15 | 15 | 20 | 16 | 91 | 9 | 6 | 153 | 7 | 10 |
| | Landslide fraction | 0.106 | 0.116 | 0.111 | 0.104 | 0.004 | 0.001 | 0.006 | 0.001 | 0.001 | 0.002 | 0.002 | 0.003 |

**Fig. 1.** Precision-recall AUC values equivalent to the ROC AUC values presented in Figure 2 of the main manuscript.

**1%**

| | Hokkaido | | | | | Nepal | | Lombok-1 | | | Lombok-2 | |
|---|---|---|---|---|---|---|---|---|---|---|---|---|
| | S068a | S046d | A116a | A018d | S085a | S019d | A157a | S156a | S032d | A129a | S156a | S032d |
| ARIA | 0.50 | 0.54 | 0.65 | 0.87 | 0.59 | 0.70 | 0.79 | 0.58 | 0.47 | 0.74 | 0.57 | 0.39 |
| Bx-S | 0.57 | 0.56 | 0.45 | 0.41 | 0.61 | 0.69 | - | 0.54 | 0.53 | 0.66 | 0.69 | 0.61 |
| PECI | 0.67 | 0.66 | 0.64 | 0.76 | 0.64 | 0.66 | 0.75 | 0.51 | 0.52 | 0.75 | 0.62 | 0.58 |
| DC_sum | 0.63 | 0.63 | 0.66 | 0.85 | 0.64 | 0.70 | 0.83 | 0.59 | 0.48 | 0.75 | 0.63 | 0.47 |
| DC_max | 0.64 | 0.64 | 0.60 | 0.85 | 0.61 | 0.71 | 0.81 | 0.60 | 0.48 | 0.76 | 0.65 | 0.49 |

**10%**

| | S068a | S046d | A116a | A018d | S085a | S019d | A157a | S156a | S032d | A129a | S156a | S032d |
|---|---|---|---|---|---|---|---|---|---|---|---|---|
| ARIA | 0.51 | 0.55 | 0.67 | 0.88 | 0.59 | 0.69 | 0.80 | 0.62 | 0.50 | 0.82 | 0.52 | 0.41 |
| Bx-S | 0.58 | 0.57 | 0.49 | 0.43 | 0.64 | 0.72 | - | 0.61 | 0.57 | 0.76 | 0.71 | 0.63 |
| PECI | 0.72 | 0.71 | 0.65 | 0.77 | 0.65 | 0.67 | 0.79 | 0.55 | 0.58 | 0.87 | 0.64 | 0.60 |
| DC_sum | 0.67 | 0.67 | 0.68 | 0.86 | 0.65 | 0.71 | 0.86 | 0.64 | 0.53 | 0.86 | 0.61 | 0.49 |
| DC_max | 0.68 | 0.69 | 0.62 | 0.87 | 0.61 | 0.70 | 0.84 | 0.65 | 0.52 | 0.86 | 0.63 | 0.51 |

**25%**

| | S068a | S046d | A116a | A018d | S085a | S019d | A157a | S156a | S032d | A129a | S156a | S032d |
|---|---|---|---|---|---|---|---|---|---|---|---|---|
| ARIA | 0.52 | 0.57 | 0.70 | 0.89 | 0.59 | 0.69 | 0.81 | 0.64 | 0.56 | 0.88 | 0.49 | 0.45 |
| Bx-S | 0.59 | 0.58 | 0.52 | 0.44 | 0.66 | 0.74 | - | 0.64 | 0.55 | 0.84 | 0.67 | 0.60 |
| PECI | 0.79 | 0.78 | 0.66 | 0.80 | 0.66 | 0.68 | 0.81 | 0.64 | 0.69 | 0.93 | 0.68 | 0.68 |
| DC_sum | 0.73 | 0.73 | 0.70 | 0.88 | 0.66 | 0.71 | 0.88 | 0.71 | 0.62 | 0.92 | 0.63 | 0.57 |
| DC_max | 0.74 | 0.75 | 0.64 | 0.89 | 0.62 | 0.71 | 0.85 | 0.72 | 0.62 | 0.92 | 0.65 | 0.59 |

**50%**

| | S068a | S046d | A116a | A018d | S085a | S019d | A157a | S156a | S032d | A129a | S156a | S032d |
|---|---|---|---|---|---|---|---|---|---|---|---|---|
| ARIA | 0.57 | 0.64 | 0.79 | 0.91 | 0.59 | 0.69 | 0.84 | - | - | - | - | - |
| Bx-S | 0.63 | 0.63 | 0.64 | 0.49 | 0.70 | 0.80 | - | - | - | - | - | - |
| PECI | 0.98 | 0.96 | 0.68 | 0.88 | 0.67 | 0.69 | 0.88 | - | - | - | - | - |
| DC_sum | 0.91 | 0.90 | 0.77 | 0.93 | 0.68 | 0.73 | 0.94 | - | - | - | - | - |
| DC_max | 0.92 | 0.92 | 0.71 | 0.95 | 0.63 | 0.73 | 0.90 | - | - | - | - | - |

| Fraction of total landslide area included in analysis | | | |
|---|---|---|---|
| Hokkaido | Nepal | Lombok-1 | Lombok-2 |
| 1% 100 | 99 | 96 | 97 |
| 10% 94 | 74 | 45 | 47 |
| 25% 73 | 39 | 14 | 15 |
| 50% 30 | 15 | 2 | 3 |

**Fig. 2.** ROC AUC values at a 200 x 220 m resolution with varying threshold between 'landslide' and 'non-landslide' pixels. (inset) the percentage of the total landslide area that is classed as 'landslide'.

**Supplement:**

Response to Reviewer 2

General comments: The authors present a quantitative comparison of multiple different methods of mapping landslides using the coherence of synthetic aperture radar images. The quantitative comparison between different methods, with tests in different regions and examination of the local features that can affect results, is particularly valuable given the ongoing work on using SAR for disaster response. The work is well written and figures are of a good standard.

There are some places where I think the assumptions underlying certain choices, both for the landslide mapping methods and the quantification of the results could be better explained; I have provide further comments on these below. There are also numerous places where I think small tweaks to the wording and slightly more explanations would be helpful to the reader.

We thank the reviewer for their helpful suggestions and have responded to each of their comments below.

Technical comments:

Quantification of different methods:

I would like to see further discussion of the use of the ROC curve to evaluate these classification methods. My understanding is that ROC curves are generally best deployed when the dataset is balanced between positive and negative examples (i.e. roughly equal numbers of landslide and non-landslide pixels), however I wonder if that is the case here, or if there are many more non-landslide pixels than landslide pixels in general?

If it's the case that the dataset is imbalanced, with more non-landslide pixels, then the ROC curve might not be the best way to assess how well your method is performing. **For example, if I take a balanced dataset then add a large number of negative examples, with their classification scores drawn from the same distribution at the existing negative examples, the false positive rate (FPR=FP/(FP+TN)) and true positive rate(TPR=TP/(TP+FN)) remain the same for a given threshold, and the ROC curve doesn't change**. However I now have many more false positives at a given threshold, meaning the precision of my classifier (=TP/(TP+FP)) will decrease (i.e. a smaller fraction of my positive classifications will be true positives). In some circumstances it could be the case that only a small fraction of my samples that are classified as positive are actually true positives, even as my TPR and FPR appear good.

Additionally, as I expand the region spanned by the SAR data, it's possible that I include more and more pixels that will have lower noise levels, e.g. as they're further away from the earthquake and so have less of the building damage, surface rupture, liquefaction etc. that can lead to false positives for landslide classification. This would lead to having a larger number of true negatives, and so an improved false positive rate, thus an improved ROC curve, but would have minimal effect on the precision, which doesn't consider true negatives. In addition to the currently presentation, you should consider presenting a precision-recall curve, or mention why the the ROC curve is preferred over the precision-recall curve. It would also be good to have you mention the fraction of each region that is covered with landslide and non-landslide pixels.

We thank the reviewer for the suggestion of using precision recall (P-R) instead of ROC. There are pros and cons to both methods and we have now checked that our conclusions are not affected by our decision to use ROC rather than P-R curves. This is because our conclusions depend on relative ranking of methods using these curves rather than absolute AUC values, and these rankings are insensitive to the choice between ROC and P-R. However, we choose to retain ROC based results within the paper. They are more appropriate than P-R for this study for two reasons

- First, true negatives are not used at all in calculation of P-R curves. Therefore P-R is only concerned with correct prediction of the smaller positive class (i.e. landslides for this study) and implicitly suggests that we are not interested in correct prediction of the larger negative class (non-landslides). Whilst this will be appropriate for many imbalanced classification problems, it is not appropriate for landslide mapping, where it is important to accurately

identify where landslides have not occurred as well as where they have occurred. This is important for correct allocation of resources in emergency response.

- Second, the example above from the reviewer (highlighted in bold) highlights that the ROC curve is essentially insensitive (or at least very weakly sensitive) to the area of negative samples (i.e. the area of the data with no landslides). This is in contrast to P-R curves, which are strongly sensitive to the area of the dataset that has no landslides. In our case, this is exactly why ROC is more appropriate than P-R and is a strong argument in favour of using ROC. The proportion of landslide pixels in each dataset varies significantly between the four events, and is dependent both on landslide density within the landslide affected area (which is event-specific) and on the area of data that is processed, i.e. how much SAR data is processed beyond the region of intense landsliding (which is an arbitrary processing decision, and could depend on the frame size, which can vary across different satellites). We wish to be able to compare performance of our classification methods across these events in a fair manner, and also do not want our performance metric to depend on arbitrary processing choices. The baseline of a precision-recall curve is dependent on the proportion of landslide pixels in the dataset, and this varies significantly between the four events (and even between different sensors, due to differences in frame size). Therefore, P-R AUC values vary a lot between events, which distracts from the main result. A classifier with no ability will consistently have an ROC AUC of 0.5 for all events, which makes this metric more suitable when carrying out analysis across several events and sensors as we have done here.

We have, however, tested using P-R instead to check that the relative *rankings* of the methods don't change for each SAR track, even if the absolute values do change and preclude useful cross-comparison.  The results are very similar, and we include these results in an appendix. If the methods are ranked for each SAR track (to avoid landslide density effects on P-R AUC), the ranking for ROC AUC and P-R AUC is identical in 46/59 of cases for the results in Figure 2 and only differ by more than 1 rank in 6 cases. The conclusions we draw from our ROC analysis are the same as those that we would draw from P-R analysis. As the fraction of landslide pixels in the validation dataset for each track represents the baseline P-R AUC, this information is included in this appendix. We also will add a short section to the main text, referring to the appendix and clarifying why ROC is more appropriate for this study.

Proposed addition to main text in section 3.5: "On all SAR tracks, there are many more landslide than non-landslide pixels. It has been suggested that for such imbalanced data, precision-recall curves can better represent classification ability than ROC AUC (Saito and Rehmsmeier, 2015). Here, we chose to use ROC analysis since precision-recall curves do not allow comparison between datasets with different proportions of landslide and non-landslide pixels and therefore between different earthquakes and SAR tracks. However, when considering the relative performance of classifiers for each track independently, we found the same conclusions could be drawn from precision-recall curves as from ROC curves. A recreation of Figure 2 using precision-recall rather than ROC AUC values can be found in Supplementary Information."

This supplementary information is supplied as Figure 1 of this review.

Histogram matching:

The use of histogram matching between the pre- and co-seismic coherence images could do with some further explanation so the reader understands the motivation and assumptions. My understanding is that this adjustment assumes that only a small number of pixels are anomalous (i.e. contain landslides)âˇAˇT otherwise you would end up removing the signal you were looking for. Furthermore, I think there is an assumption that the coherence of the pixels in the image that's adjusted have all been affected in the same way. For example, if only the southern part of the image had been covered in snow between the final pre-seismic and first post-seismic SAR images, then adjusting the entire coseismic coherence image based on a simple matching of histograms would not be the correct approach, however if the second coherence image just had a longer temporal baseline then the extra temporal decorrelation might be removable by histogram matching.

Yes that is correct. We will clarify this in the text.

Proposed new text at line 167: "This step is done to account for different levels of temporal decorrelation when the pre-event and co-event interferograms have different temporal baselines. It assumes only a small fraction of the pixels are affected by landslides so that the landslide signal is not removed from the co-event interferogram."

Optimum thresholds:

It might be helpful for the discussion to go into more detail on what the optimum threshold for flagging landslides would be for each method, as this is what would be required before use by a first responder. Currently the presentation of your results is threshold free (i.e. in terms of the ROC curve), apart from when you make plots. The discussion around line 300-305 explains that you choose a threshold for plotting in order to be consistent with the number of points plotted in the landslide area plot in Fig 4, but it's not clear if this would be a useful threshold for response. The comparison you chose to present in figure 4 was also slightly confusing to me; in line 236 you say that you're only classifying pixels as 'landslide' pixels if they have >25% area of landslides, however for the plot in figure 4 you plot pixels that have >1% area of landslides, and then choose the classification threshold based on the number of pixels that have >1% landslide area. Is there a reason for the different threshold of landslide area? Would it not be better to choose the classification threshold from some desired trade off between true positive and false positive rates on the ROC curve (or a trade off between precision and recall)?

This comment relates to three different thresholds: 1) the minimum landslide area density below which landslide density is not displayed in Figures 3 and 4; 2) the minimum classifier value below which the classification surface is not displayed in Figures 3 and 4; and 3) the minimum landslide area density below which a pixel is not considered 'landslide affected' in ROC and P-R performance evaluation. We have chosen to approach these differently as they have different purposes.

The first two are 'display thresholds', and as the reviewer points out this would not necessarily be a useful threshold for response, in fact, establishing the threshold used in (2) would not be possible since the landslide density would be unknown. However, these 'display thresholds' affect the look of the figures rather than our findings. In the 'ground truth' maps of observed landslide density (Figures 3g and 4b, d, f) we chose to plot landslide areal density and to mask pixels with landslide area density <1%. This threshold of 1% was chosen because it captures the majority of the landslide-affected pixels. We chose the thresholds for the coherence-based surfaces such that the masked area in each would be approximately equal to that in the observed landslide density map. We agree, we have not explored this choice of threshold here and whether it would be suitable for emergency response. Since the figure shows ALOS-2 data and the time between image acquisitions for these data varies between events, which affects the coherence, we did not expect to be able to extract a threshold that could be applied in future events at this stage. Choosing the classification threshold based on a trade-off between true positive rate and false positive rate, or precision and recall would result in differences between events in the figures that might not fairly represent how well they were modelled. We will add the following text to Section 5.3 (Future Work) in the Discussion.

"Here we assessed classifier performance using ROC analysis, which does not require a threshold to be applied to the classification surface. However, if SAR methods are to be applied to future events for emergency response, it will be necessary to set a threshold between 'landslide' and 'non-landslide'. Here the time between image acquisitions varied significantly between events, making it unlikely that a threshold could be selected that would work well for either Sentinel-1 or ALOS-2 across all events. However, this may be possible in the future when more events have been studied and SAR data with more regular acquisitions are available. Further work is therefore needed to establish such thresholds, which will be determined according to the requirements of emergency responders and their relative tolerance for false positives and false negatives."

The final threshold is required to coarsen the resolution of the landslide observations to match that of the classifiers. In this case, we chose to set the threshold higher than the 1% threshold used in Figures 3 and 4 and considered only cells with >25% landslide area to be 'landslide affected' for the

purpose of ROC and precision-recall analysis. Pixels containing more landslides are more important to identify, for example because the chance that a road within the pixel has been blocked by a landslide is higher. But if the threshold is set too high, not enough pixels are classed as 'landslide' to be able to carry out meaningful ROC or precision-recall analysis, especially in Lombok where the landslide density was lowest, and many regions of landsliding will be missed by the classifier . The choice of 25% is therefore somewhat arbitrary but represents a trade-off between reliability of the ROC and P-R analysis and sensitivity to pixels less affected by landslides.

We have also tested at other thresholds (1%, 10% and 50%). These results will be included in as an additional figure in Supplementary Information (attached as Figure 2 of this review) and referred to in the main text. In general, more severely impacted pixels have a stronger signal in the coherence surfaces and so are easier to detect. Therefore, increasing this threshold (e.g. to 50%) produces a higher ROC AUC. The difference in ROC AUC between setting this threshold at 1% and 25% can be up to 0.18, but in most cases was less than 0.1. Again, this choice makes little difference to the relative ranking of the methods and thus does not influence our conclusions.

'ARIA' method naming:

The use of the descriptor 'ARIA' method isn't standard to my knowledge, and may be misleading. JPL's ARIA Project does employ this method, although it is also working on many alternative approaches. Furthermore, the method, or similar variants of it, has been used by many groups (e.g. Fieldling et al (2005)) and is not referred to as the 'ARIA' method in these publications. One term I have seen applied is 'Coherence Change Detection' (e.g. Washaya et al. 2018), however this may not be the most useful, as all of your methods are change detection using coherence. Perhaps 'coherence loss' would be a better term?

Citations for the paragraph above:

- Fielding, E. J. et al. (2005) 'Surface ruptures and building damage of the 2003Bam, Iran, earthquake mapped by satellite synthetic aperture radar interferometric correlation', Journal of Geophysical Research: Solid Earth, 110(3), pp. 1–15. doi:10.1029/2004JB003299.

- Washaya, P., Balz, T. and Mohamadi, B. (2018) 'Coherence Change-Detection with Sentinel-1 for Natural and Anthropogenic Disaster Monitoring in Urban Areas', Remote Sensing, 10(7), p. 1026. doi: 10.3390/rs10071026.

Thank you for drawing our attention to these papers, we will add these to the text and include a more complete description of damage-detection methods based on coherence loss. To match the method we refer to as "post-event coherence increase" (PECI), we will alter the text and refer to the ARIA method as "co-event coherence loss" (CECL)

Line by Line comments:

Line 73 - ï¿£'The signal-to-noise ratio of each pixel in an InSAR image is described by its coherence'. This statement feels incomplete to meâ˘A˘Tin your work the 'signal' is the decorrelation of the pixel, and in circumstances where the phase is the signal (e.g. deformation time series), the decorrelation is only one contributor to the noise (e.g. can also have atmospheric and ionospheric noise which doesn't affect the coherence).

By 'signal to noise', we meant the signal quality of the interferogram when it is used for ground deformation studies, not the signal in coherence studies, and were referring to high-frequency noise, not the longer wavelength nuisance signals that arise from ionospheric or tropospheric phase changes. We will reword this statement to make it clearer.

Proposed new text: "When using an interferogram to map ground deformation, it is important that the signals recorded at a given location in the two SAR images are correlated, as decorrelation will result in high-frequency noise. In order to assess this and to identify noisy pixels, the coherence γ is estimated for every pixel from the similarity in the two SAR images in amplitude and phase difference, for a small ensemble of $n$ pixels (Eq. 1, Just and Bamler, 1994):"

Line 77 - I think some mention of the assumptions of the boxcar and sibling methods would be useful. My understanding is that the box-car method assumes that the surrounding pixels are statistically similar, and the sibling method identifies pixels that are similar through time (based on the amplitude) and assumes that these remain similar for subsequent acquisitions. I note that the sibling method is briefly explained on line 212, but more information in the introduction I feel would be helpful to the reader.

Yes this is correct, we will add more information here for clarity

Proposed new text: "The ensemble is chosen so that the pixels used in the calculation are expected to be similar. In a 'boxcar' method, it is assumed that pixels immediately adjacent to the target pixel are similar to it (e.g. Hansen, 2001; Yun et al. 2015). In a 'sibling' method an assessment is carried out for every pixel to identify pixels that are statistically similar to it. For example, the sibling method of Spaans and Hooper identifies pixels that have similar amplitude behaviour through time."

Line 87 - perhaps include a definition of perpendicular baseline, and mention that orbital controls on modern satellites have rendered this a much smaller issue?

Thank you, we will add this to the text.

Proposed new text: "When the perpendicular baseline (the distance between the locations at which the satellite acquired the two SAR images perpendicular to the flight and look directions) of the SAR image pair used to form an interferogram is sufficiently small, this spatial component will be small compared to any temporal decorrelation (Zebker and Villasenor, 1992). For modern satellites, this will be the case most of the time."

Line 100 - add citation on the effect of vegetation on coherence at different wavelengths?

We will cite Zebker and Villasenor (1992) here, who compare decorrelation due to movement of scatterers at L-band and C-band wavelengths.

Line 106 'as it is impossible to combine data from both tracks' - it would be helpful to clarify what is meant by 'combine' (i.e. you can't calculate the coherence between different tracks)

Thank you for identifying this, we will change this to:

"as it is impossible to calculate a combined coherence surface using data from two tracks"

Line 119 - what is meant by 'high resolution' data for ALOS-2? Might be helpful to mention the acquisition mode for clarity?

We will specify in the text that these data were acquired in "stripmap" mode at a resolution of 3 – 10 m. The resolutions of each track can be found in Table 1.

Line 165 - need to clarify that the user must choose a coherence loss threshold for flagging damage, the current sentence structure implies that all pixels that have any coherence decrease at all are flagged as damaged

Here we apply this method to landslides and do not apply a threshold for the ROC analysis, but in the original Yun et al. 2015 paper, they use 0 as the threshold, so that any pixel whose coherence decreases is flagged as damaged. We will clarify this in the text.

Line 179 - I think it's worth reminding the reader here that the sibling pixels have been specifically identified to be behaving similarly before the earthquake by looking at the amplitude time series, and it's on that basis that they're expected to behave similarly

Yes, that's a good suggestion, we will add this to the text.

Proposed new text: "For every pixel, an ensemble of 'siblings' is selected that have similar behaviour in terms of amplitude in a time series of pre-event imagery."

Line 185 - post-event coherence - make it clear that this coherence is calculated using the boxcar method (which I assume it is?)

This was stated at line 176, but we will alter the text to ensure it is clear that the boxcar coherence is used in section 3.3.1, 3.3.3, 3.3.4, and 3.3.5.

Line 233 - 'If over 95% of an aggregate pixel is made up of masked pixels, the aggregate pixel was masked' - how do you choose this number? Are you results sensitive to this choice? What fraction of pixels end up being masked out?

The effect of this choice on ROC AUC is very low (< 0.02). However, altering this can significantly alter the number of pixels masked, for example on track 19 in Nepal, altering this threshold from 95% to 5% decreases the number of aggregate pixels used in the analysis from 246,375 to 148,829. The effect was much weaker in Hokkaido and Lombok, where fewer pixels were masked due to distortion on steep slopes. For example, in Hokkaido on track 68, altering the threshold from 95% to 5% decreases the number of aggregate pixels from 19,754 to 19,235. We chose to set the threshold high to avoid masking any landslides unnecessarily and to therefore maintain good spatial coverage for our classification surface.

Additonal text at line 234: "This high threshold of 95% was chosen to minimise the loss of spatial coverage due to the masks. Varying the threshold between 95% and 5% had little difference in terms of the number of pixels used in the analysis in Hokkaido and Lombok (<5%), but in Nepal, where more pixels were masked due to distortion on steep slopes, decreasing the threshold to 5% resulted in a loss of coverage of around 40% on S085a. Moving this threshold made very little difference to the results presented in Section 4.1."

Line 234 - ï´z£'In this study, we did not attempt to map SAR classification surface values directly to landslide areal density values, as this has not been attempted in previous studies and may not be possible' - would be good to get more info about why this is/ isn't possible, and a citation for the previous work. Is this comment based on the work of Burrows et al. (2019)?

All of the studies done so far applying SAR data to landslide detection have used a binary validation landslide dataset and have not attempted to recreate landslide areal density (Aimaiti et al. 2019; Burrows et al. 2019; Ge et al. 2020; Jung and Yun 2019; Masato et al. 2020; Yun et al. 2015). Although it may be possible to extract landslide areal density from SAR data, this is uncertain as differences in viewing geometry, land cover type and (particularly with the ALOS-2 data used here) differences in the temporal baselines of the coherence maps may mean that there is too much variation between events to establish rules on this. This will be clarified in the text.

Line 236 - how did you select the 25% threshold? How much does this choice affect your results? How large is this area, and what fraction of the total landslide area is missed by virtue of the fact that it falls below this threshold? Would be helpful to have these questions discussed in the text, particularly if this choice has a large affect on AUC values.

The choice of this threshold has been explained in our response to the 'Optimum thresholds' comment in this review. The effect of this choice will be demonstrated in Supplementary material where we varied this threshold between 1% and 50%. In general, setting the threshold high results in slightly higher ROC AUC because the signal of an aggregate pixel containing more landslide pixels is stronger but the relative performance of different classifiers remains the same.

For a 200 x 220 m pixel, 25% corresponds to an area of 11,000 m2 out of the total 44,000 m2 area of each aggregate pixel. Applying this threshold results in around 27% of the total landslide area being classified as 'non-landslide' in Hokkaido, around 61% in Nepal and in Lombok, where the average landslide size is comparably small, around 86% for Lombok-1 and 85% for Lombok-2. When the threshold is dropped to 10%, the total excluded landslide area decreases to 6% in Hokkaido, 26% in Nepal and 55% and 53% in Lombok-1 and -2 respectively. Information on the proportion of excluded pixels with the threshold set at 1%, 10%, 25% and 50% will be included in the Supplementary Information referred to above (Figure 2 of this response).

Line 277 - 'ï z£the 6-day Sentinel-1 acquisition window' - this is a little unclear, to my mind 'window' implies that data was acquired at some point in that time range. Rather than 'window' could say 'acquisition frequency'?

We will change this to repeat time to avoid confusion.

Line 287 - for clarity it may be worth reminding the reader that the L-band image is from the ALOS 2 satellite

We will add this to the text

Line 324 - a brief discussion of what is meant by 'higher quality siblings' would be helpful (I imagine that they're more statistically similar?)

If siblings are selected using SAR images acquired more recently before the earthquake, then there is less time for pixels to have been altered by changes to the ground surface, and their behaviour is more likely to remain similar in the co-seismic image (assuming there are no landslides)

Proposed revised text: "As Sentinel-1 imagery is acquired every 12 days, more images were available for this calculation and were acquired over a shorter time period. This allows less time for pixels to be altered by changes to the ground surface, meaning that, for non-landslide pixels, a pixel and its siblings are likely to be more similar. In this way, the siblings selected by RapidSAR for Sentinel-1 imagery may have been of a higher quality than those for ALOS-2, giving a more reliable coherence estimate."

Line 425 - You mention rivers giving false positives for the Bx-S method. Would it be possible to identify rivers by applying the Bx-S method to a pre-seismic coherence image (using the same pixel siblings) and then use that as a mask on your co-seismic landslide image? You could propose this if so.

Yes this might be possible and could be a good way to remove these false positives. Currently, we do not mask rivers since places where landslides and rivers intersect are particularly dangerous due to the potential for landslide dams and a map of predicted landslide locations with rivers masked might lead to this hazard being missed.

This is related to a comment made by the other reviewer: "You could mask the rivers. Since the single landslide is not the target of your approach, the detection of a single landslide dam should not be possible and the river-mask duable." We propose the following addition to the text to address both of these comments.
Proposed addition to text: "A variety of methods could be used to identify and remove rivers from our analysis (including using a pre-event Bx-S surface). However, since areas where landslides and rivers intersect are particularly hazardous due to the potential for landslide dams and associated flash flooding, we did not mask rivers in this study. We suggest that any product based on SAR coherence supplied to emergency response coordinators should have rivers overlaid. This would both mask false positives due to rivers and allow identification of locations where rivers pass through areas of intense landsliding."

Line 469 - ï z£ 'The ARIA method is the best performing method when only one L-band image is available' - does this mean 'one post-seismic L-band SAR image'? I think this could be clarified, as I could read this to mean that only on L-band SAR image is available in total.

Thank you for pointing this out, we will change this to "one post-event L-band image".

Figure 6 - how is the threshold for these plots chosen? Could you clarify in the caption?

The 10% of pixels most likely to be landslides are plotted. This will be added to the Figure caption.

Technical Corrections: For words in quotation marks apostrophes are being used throughout, rather than opening and closing quotation marks. Please adjust.

Thank you for pointing this out, we will correct this.

Line 185 - double use of 'by a landslide' in this sentence.

Thank you, this has now been corrected

Line 235 and 443 - add a comma after 'Thus' (consistent with 'Thus' on line 10)

Thank you, we will adjust this for consistency

Table 1 - 'x' has inconsistent formatting or is missing in some cases

Thank you for bringing this to our attention, we will correct this table.

---

## Author Response (AR1)

**Authors' response to comments**

We would like to thank the two reviewers for their comments, which have improved the manuscript greatly. Here we provide a complete response to comments from both reviewers, along with the revised manuscript in which all changes are indicated.

**Reviewer 1**

**Comment 1**

Line 5 "triggering events" You can specify the type of events

**Manuscript Change**: We have changed this to "*triggering earthquakes*" (Line 5, revised manuscript)

**Comment 2**

Line 6 "ARIA", specify the acronym

**Response**: We are renaming this method based on comments from reviewer 2 so this will be changed to "co-event coherence loss method"

**Manuscript change**: "*the ARIA method performs best*" at line 6 of original manuscript has been changed to "*co-event coherence loss (CECL) is the best performing method*" at line 6 of the revised manuscript.

**Comment 3**

Line 10 "useful landslide density", I would say landslide mapping

**Response**: We used landslide density so that readers do not expect individual mapped landslides, which is the most common output from mapping landslides using optical satellite imagery.

**Manuscript change**: none

**Comment 4**

Line 22 "could have been much longer had the earthquake occurred during Nepal's monsoon season", check this

**Response:** This statement is supported by Robinson et al. 2019, which is referenced earlier in the same sentence

**Manuscript Change**: none

**Comment 5**

Line 30 "reliability of global empirical rainfall-triggered landslide susceptibility maps", the scale itself is a limitation

**Response**: The spatial scale of the empirical models may be a disadvantage for some applications, but immediately after the earthquake, low resolution information over a large spatial scale is useful for giving an overview of severely impacted areas to emergency response coordinators (Discussed in Williams et al. *NHESS*. 2018). Although our analysis is carried out at a slightly higher resolution than these empirical models (200 x 220 m pixels as opposed to 1 km2, which is used by Nowicki-Jessee), the improvement in resolution in this study is fairly limited.

**Manuscript Change**: Text added at line 26 of revised manuscript: "*The outputs of these models have a comparatively low spatial resolution (around 1 km2 in the case of Nowicki Jessee et al. 2018), but can be used to provide an overview of the most severely impacted areas*"

**Comment 6**

Line 31 "style of landsliding,", I would say landslide type

**Manuscript Change:** "*style of landsliding*" at line 31 of original manuscript changed to "*type of landslides*" in line 33 of revised manuscript

**Comment 7**

Line 41 there are thousands of published paper in this field. I think the reference list can be extended, see some recent examples: _Dai, K., Li, Z., Tomás, R., Liu, G., Yu, B., Wang, X., ... & Stockamp, J. (2016). Monitoring activity at the Daguangbao mega-landslide (China) using Sentinel-1 TOPS time series interferometry. Remote Sensing of Environment, 186, 501-513. _Solari, L., Del Soldato, M., Raspini, F., Barra, A., Bianchini, S., Confuorto, P., ... & Crosetto, M. (2020). Review of Satellite Interferometry for Landslide Detection in Italy. Remote Sensing, 12(8), 1351. _Aslan, G., Foumelis, M., Raucoules, D., De Michele, M., Bernardie, S., & Cakir, Z. (2020). Landslide Mapping and Monitoring Using Persistent Scatterer Interferometry (PSI) Technique in the French Alps. Remote Sensing, 12(8), 1305. _Reyes-Carmona, C., Barra, A., Galve, J. P., Monserrat, O., Pérez-Peña, J. V., Mateos, R. M., ... & Azañón, J. M. (2020). Sentinel-1 DInSAR for Monitoring Active Landslides in Critical Infrastructures: The Case of the Rules Reservoir (Southern Spain). Remote Sensing, 12(5), 809. _Hu, X., Bürgmann, R., Lu, Z., Handwerger, A. L., Wang, T., & Miao, R. (2019). Mobility, thickness, and hydraulic diffusivity of the slow˘AR˘moving Monroe landslide in California revealed by L⸣˘AR˘ band satellite radar interferometry. Journal of Geophysical Research: Solid Earth, 124(7), 7504-7518.

**Response:** Thank you for drawing our attention to these materials, these references have now been added

**Manuscript Change**: References at line 41 (original manuscript) "*e.g. Boni et al, 2018; Handwerger et al., 2019*" changed to "*e.g. Aslan et al., 2020; Bonì et al., 2018; Dai et al., 2016; Handwerger et al., 2019; Hu et al., 2019; Reyes-Carmona et al., 2020; Solari et al., 2020)*" at line 43 of the revised manuscript. Aslan et al., (2020); Dai et al., 2016; Hu et al., 2019; Reyes-Carmona et al., 2020; Solari et al., 2020 have been added to the bibliography.

**Comment 8**

Line 68 "method"–> technique

**Manuscript Change:** "*method*" at line 68 of original manuscript changed to "*technique*" at line 71 of the revised manuscript.

**Comment 9**

Line 77 any reference for the Boxcar method?

**Response:** This is the most common method of calculating coherence, we have altered the text to make this clearer and added some examples of texts that use this method.

**Manuscript Change:** Line 77 of original manuscript "*The ensemble of pixels used in the calculation can be selected either using a 'boxcar' method, in which a small rectangular ensemble of pixels immediately adjacent to and centred on the target pixel are used…*" changed to "*The ensemble is chosen so that the pixels used in the calculation are expected to be similar. In a "boxcar" method, it is assumed that pixels immediately adjacent to and centred on the target pixel are similar to it (e.g.
Hanssen, 2001; Yun et al., 2015).*" at line 81 of the revised manuscript. A reference for Hansen, 2001 has been added to the bibliography.

**Comment 10**

Line 131 "landslides were triggered by the earthquake", the occurrence of a typhoon the day before is certainly a trigger as well.

**Response**: We agree that the typhoon the day before the earthquake will have caused widespread pore-pressure increase across the study area and that this may have played an important role in the initiation of some of the observed landslides. However, the landslides have been widely attributed to the earthquake as the primary trigger, with debate over the role of rainfall being focussed on the extent to which it pre-conditioned the slopes for failure (e.g. Osanai et al., 2019; Wang et al., 2019; Yamagishi and Yamazaki, 2018; Zhang et al., 2019). We are interested in this event because of the close sequencing of intense rain and shaking rather than as a test of our ability to distinguish the different triggers. We have removed reference to the earthquake as the trigger since this is unnecessary and replaced this with information on how the landslide data were collected.

**Manuscript Change**: "*with the advantage that because the landslides were triggered by the earthquake, we know more precisely when they occurred.*" at line 130 of the original manuscript changed to "*with the advantage that because the typhoon and earthquake occurred one day apart, and aerial imagery of the triggered landslides was acquired immediately afterwards, we know more precisely when the landslides occurred (Yamagishi and Yamazaki, 2018).*" at line 143 of the revised manuscript. A reference for Yamagishi and Yamazaki (2018) has been added to the bibliography.

**Comment 11**

L140 and followings, what about the type of landslides that have been triggered?

**Response**: Thank you for identifying this omission, we will add information on landslide type.

**Manuscript Change:** Text added at line 127 of the revised manuscript. "*Finally, the type of landslides triggered by our four case study earthquakes were typical of landslides triggered by earthquakes (Keefer, 1984). The majority of ground failures in the four earthquakes were slides. In Nepal, ground failures were primarily a mixture of slides and falls, with the exception of a large debris avalanche in the Langtang Valley. For all four earthquakes, failure surfaces were at shallow depths in most cases with a small number of exceptions (Collins and Jibson, 2015; Ferrario, 2019; Yamagishi and Yamazaki, 2018).*" References for Keefer (1984) and Yamagishi and Yamazaki (2018) have been added to the bibliography.

**Comment 12**

Fig.1 is the inset in (c) Lombok 2? You should specify this in the caption

**Response**: Thank you for identifying this omission, we will add this to the figure caption

**Manuscript Change**: The Figure caption for Figure 1 has been changed from "*Figure 1. (a, b, c) Landslides triggered by the 2015 Gorkha, Nepal earthquake, the 2018 Hokkaido, Japan earthquake, and the 2018 Lombok, Indonesia earthquakes, respectively, plotted as the areal density of landsliding based on 1 km2 cells (Roback et al., 2018; Zhang et al., 2019; Ferrario, 2019). ALOS-2 SAR acquisitions are shown in green and Sentinel-1 in blue. (d, e, f) Acquisition dates and track numbers of SAR imagery used in this study. Earthquakes are shown as red vertical lines, black symbols show pre-event image acquisition dates, and red symbols show post-event dates.*" to "*Figure 1. (a, b, c) Landslides triggered by the 2015 Gorkha, Nepal earthquake, the 2018 Hokkaido,*

*Japan earthquake, and the 2018 Lombok, Indonesia earthquakes, respectively, plotted as the areal density of landsliding based on 1 km2 cells (Roback et al., 2018; Zhang et al., 2019; Ferrario, 2019). ALOS-2 SAR acquisitions are shown in green and Sentinel-1 in blue. In (c), a second earthquake, referred to as "Lombok-2" is inset. (d, e, f) Acquisition dates and track numbers of SAR imagery used in this study. Earthquakes are shown as red vertical lines, black symbols show pre-event image acquisition dates, and red symbols show post-event dates."*

**Comment 13**

L185 check the double "by a landslide"

**Manuscript Change**: Second *"by a landslide"* removed at line 206 of the revised manuscript.

**Comment:** L379 and one day with images from ROSE-L

**Response:** Thank you, I was not aware of this satellite and have added a reference to this mission

**Manuscript Change:** "*In the future, however, the availability of L-band SAR is likely to increase with the planned NiSAR satellite constellation, which will acquire data continuously with a 12 day repeat time over all landmasses globally (Sharma, 2019).*" at line 377 of original manuscript changed to "*In the future, however, the availability of L-band SAR is likely to increase with the planned NISAR and ROSE-L satellite constellations. NISAR, a joint NASA-ISRO mission planned to launch in 2022, will acquire L-band data continuously with a 12-day repeat time over all landmasses globally (Sharma, 2019), while the ESA ROSE-L satellite, whose launch date is planned in 2026, will have 6-day global repeat coverage, and 3-day in Europe (Pierdicca et al., 2019).*" at line 414 of the revised manuscript. A reference for Pierdicca et al (2019) has been added to the bibliography.

**Comment 14**

L398 do you have an estimate of the number/percentage of false positives due to the buildings? The same comment for the number of false positives due to wind or snow.

**Response**: It is difficult to quantify the number of false positives resulting from a specific source as we do not threshold the classifiers in our analysis. However, we have tried to assess the impact of buildings by repeating the ROC analysis with urban areas masked based on the ESA CCI landcover map. We found very little change in ROC AUC (<0.02), as these land cover types make up a relatively small part of the study area.

Quantifying false positives in snow-covered areas is difficult, since these are also likely to be underrepresented in the landslide inventory generated from optical satellite imagery (Roback et al. 2018). Furthermore, the dependence of snow cover on season, elevation and climate means that any measure we obtain of false positives due to snow is unlikely to be applicable in future events.

Quantification of false positives due to wind damage is also difficult as there are no reliable validation data available for this. From visual inspection, it appeared that the signal was strongest for areas mapped as "evergreen needleleaf forest" but masking these areas did not noticeably improve ROC AUC (<0.02).

It is difficult to predict how these false positives might affect the applicability of SAR data to future events, as this will depend on the extent of urban development (in the case of building damage), the climate, elevation and season (in the case of snow) and on the occurrence of storms (in the case of wind damage). Therefore, while we note that these are potential causes of error, we do not think it is useful to try to quantify these further.

**Manuscript change**: none

**Comment 15**

You could mask the rivers. Since the single landslide is not the target of your approach, the detection of a single landslide dam should not be possible and the river-mask duable.

**Response:** We agree this would be possible, but although we cannot map individual landslide dams, points where rivers pass through areas of high landslide density can be used to identify locations where a landslide dam is more probable (e.g. Robinson et al. *BSSA*, 2018) and therefore which are worthy of further investigation. Therefore, although we acknowledge that a landslide signal from SAR data is less reliable in a river than elsewhere, masking them would lead to a more confusing product where landslides are not allowed to intersect with rivers. Instead we suggest that rivers should be drawn over the coherence products. In response to the comments in this review and the following comment made by the other reviewer "*Line 425 - You mention rivers giving false positives for the Bx-S method. Would it be possible to identify rivers by applying the Bx-S method to a pre-seismic coherence image (using the same pixel siblings) and then use that as a mask on your co-seismic landslide image? You could propose this if so.",* we will add the following sentences to the manuscript:

**Manuscript Change:** "*A variety of methods could be used to identify and remove rivers from our analysis (including using a pre-event Bx-S surface). However, since areas where landslides and rivers intersect are particularly hazardous due to the potential for landslide dams and associated flash flooding, we did not mask rivers in this study. We suggest that any product based on SAR coherence supplied to emergency response*

*coordinators should have rivers overlaid. This would both mask false positives due to rivers and allow identification of locations where rivers pass through areas of intense landsliding*." added at line 469 of revised manuscript.

**Reviewer 2**

**Comment 1**

Quantification of different methods:

I would like to see further discussion of the use of the ROC curve to evaluate these classification methods. My understanding is that ROC curves are generally best deployed when the dataset is balanced between positive and negative examples (i.e. roughly equal numbers of landslide and non-landslide pixels), however I wonder if that is the case here, or if there are many more non-landslide pixels than landslide pixels in general?

If it's the case that the dataset is imbalanced, with more non-landslide pixels, then the ROC curve might not be the best way to assess how well your method is performing. **For example, if I take a balanced dataset then add a large number of negative examples, with their classification scores drawn from the same distribution at the existing negative examples, the false positive rate (FPR=FP/(FP+TN)) and true positive rate(TPR=TP/(TP+FN)) remain the same for a given threshold, and the ROC curve doesn't change**. However I now have many more false positives at a given threshold, meaning the precision of my classifier (=TP/(TP+FP)) will decrease (i.e. a smaller fraction of my positive classifications will be true positives). In some circumstances it could be the case that only a small fraction of my samples that are classified as positive are actually true positives, even as my TPR and FPR appear good.

Additionally, as I expand the region spanned by the SAR data, it's possible that I include more and more pixels that will have lower noise levels, e.g. as they're further away from the earthquake and so have less of the building damage, surface rupture, liquefaction etc. that can lead to false positives for landslide classification. This would lead to having a larger number of true negatives, and so an improved false positive rate, thus an improved ROC curve, but would have minimal effect on the precision, which doesn't consider true negatives.

In addition to the currently presentation, you should consider presenting a precision-recall curve, or mention why the the ROC curve is preferred over the precision-recall curve. It would also be good to have you mention the fraction of each region that is covered with landslide and non-landslide pixels.

**Response:** We thank the reviewer for the suggestion of using precision recall (P-R) instead of ROC. There are pros and cons to both methods and we have now checked that our conclusions are not affected by our decision to use ROC rather than P-R curves. This is because our conclusions depend on relative ranking of methods using these curves rather than absolute AUC values, and these rankings are insensitive to the choice between ROC and P-R. However, we choose to retain ROC based results within the paper. They are more appropriate than P-R for this study for two reasons

• First, true negatives are not used at all in calculation of P-R curves. Therefore P-R is only concerned with correct prediction of the smaller positive class (i.e. landslides for this study) and implicitly suggests that we are not interested in correct prediction of the larger negative class (non-landslides). Whilst this will be appropriate for many imbalanced classification problems, it is not appropriate for landslide mapping, where it is important to accurately identify where landslides have not occurred as well as where they have occurred. This is important for correct allocation of resources in emergency response.

• Second, the example above from the reviewer (highlighted in bold) highlights that the ROC curve is essentially insensitive (or at least very weakly sensitive) to the size of the area of negative samples (i.e. the area of the data with no landslides). This is in contrast to P-R curves, which are strongly sensitive to the size of the area of the dataset that has no landslides. In our case, this is exactly why ROC is more appropriate than P-R and is a strong argument in favour of using ROC. The proportion of landslide pixels in each dataset varies significantly between the four events, and is dependent both on landslide density within the landslide affected area (which is event-specific) and on the area of data that is processed, i.e. how much SAR data is processed beyond the region of intense landsliding (which is an arbitrary processing decision, and could depend on the frame size, which can vary across different satellites). We wish to be able to compare performance of our classification methods across these events in a fair manner, and also do not want our performance metric to depend on arbitrary processing choices. The baseline of a precision-recall curve is dependent on the proportion of landslide pixels in the dataset, and this varies significantly between the four events (and even between different sensors, due to differences in frame size). Therefore, P-R AUC values vary a lot between events, which distracts from the main result. A classifier with no ability will consistently have an ROC AUC of 0.5 for all events, which makes this metric more suitable when carrying out systematic analysis across several events and sensors as we have done here.

We have, however, tested using P-R instead to check that the relative *rankings* of the methods don't change for each SAR track, even if the absolute values do change and preclude useful cross-comparison. The results are very similar, and we include these results in an appendix. If the methods are ranked for each SAR track (to avoid landslide density effects on P-R AUC), the ranking for ROC AUC and P-R AUC is identical in 46/59 of cases for the results in Figure 2 and only differ by more than 1 rank in 6 cases. The conclusions we draw from our ROC analysis are the same as those that we would draw from P-R analysis. As the fraction of landslide pixels in the validation dataset for each track represents the baseline P-R AUC, this information is included in this appendix. We have also added a short section to the main text, referring to the appendix and clarifying why ROC is more appropriate for this study.

**Manuscript Change:** text added at line 285 of revised manuscript: *"On all SAR tracks, there are many more non-landslide than landslide pixels. It has been suggested that for such imbalanced data, precision-recall curves can better represent classification ability than ROC AUC (Saito and Rehmsmeier, 2015). Here, we chose to use ROC analysis since precision-recall curves do not allow comparison between datasets with different proportions of landslide and non-landslide pixels and therefore between different earthquakes and SAR tracks. However, when considering the relative performance of classifiers for each track independently, we found the same conclusions could be drawn from precision-recall curves as from ROC curves. A recreation of Figure 2a using precision-recall rather than ROC AUC values can be found in Supplementary Information."* A reference for Saito and Rehmseier (2015) has been added to the bibliography.

**Comment 2**

Histogram matching:

The use of histogram matching between the pre- and co-seismic coherence images could do with some further explanation so the reader understands the motivation and assumptions. My understanding is that this adjustment assumes that only a small number of pixels are anomalous (i.e. contain landslides)a˘A˘T otherwise you would end up removing the signal you were looking for. Furthermore, I think there is an assumption that the coherence of the pixels in the image that's adjusted have all been affected in the same way. For example, if only the southern part of the image had been covered in snow between the final pre-seismic and first post-seismic SAR images, then adjusting the entire coseismic coherence image based on a simple matching of histograms would not be the correct approach, however if the second coherence image just had a longer temporal baseline then the extra temporal decorrelation might be removable by histogram matching.

**Response:** Yes that is correct. We will clarify this in the text.

**Manuscript Change:** *"This process accounts for different levels of bulk temporal decorrelation in the pre-event and co-event interferograms. It assumes only a small fraction of the pixels are affected by landslides so that the landslide signal is not removed from the co-event interferogram."* added at line 184 of revised manuscript.

**Comment 3**

Optimum thresholds:

It might be helpful for the discussion to go into more detail on what the optimum threshold for flagging landslides would be for each method, as this is what would be required before use by a first responder. Currently the presentation of your results is threshold free (i.e. in terms of the ROC curve), apart from when you make plots. The discussion around line 300-305 explains that you choose a threshold for plotting in order to be consistent with the number of points plotted in the landslide area plot in Fig 4, but it's not clear if this would be a useful threshold for response. The comparison you chose to present in figure 4 was also slightly confusing to me; in line 236 you say that you're only classifying pixels as 'landslide' pixels if they have >25% area of landslides, however for the plot in figure 4 you plot pixels that have >1% area of landslides, and then choose the classification threshold based on the number of pixels that have >1% landslide area. Is there a reason for the different threshold of landslide area? Would it not be better to choose the classification threshold from some desired trade off between true positive and false positive rates on the ROC curve (or a trade off between precision and recall)?

**Response:**

This comment relates to three different thresholds: 1) the minimum landslide area density below which landslide density is not displayed in Figures 3 and 4; 2) the minimum classifier value below which the classification surface is not displayed in Figures 3 and 4; and 3) the minimum landslide area density below which a pixel is not considered 'landslide affected' in ROC and P-R performance evaluation. We have chosen to approach these differently as they have different purposes.

The first two are 'display thresholds', and as the reviewer points out this would not necessarily be a useful threshold for response, in fact, establishing the threshold used in (2) would not be possible since the landslide density would be unknown. However, these 'display thresholds' only affect the look of the figures rather than our findings, and are selected simply to graphically illustrate our results in an appropriate way. In the 'ground

truth' maps of observed landslide density (Figures 3g and 4b, d, f) we chose to plot landslide areal density and to mask pixels with landslide area density <1%. This threshold of 1% was chosen because it captures the majority of the landslide-affected pixels. We chose the thresholds for the coherence-based surfaces such that the masked area in each would be approximately equal to that in the observed landslide density map. We agree, we have not explored this choice of threshold here and whether it would be suitable for emergency response. Since the figure shows ALOS-2 data and the time between image acquisitions for these data varies between events, which affects the coherence, we did not expect to be able to extract a threshold that could be applied in future events at this stage. Choosing the classification threshold based on a trade-off between true positive rate and false positive rate, or precision and recall would result in differences between events in the figures that might not fairly represent how well they were modelled. We have added text to Section 5.3 (Future Work) in the Discussion to make this clear.

The final threshold is required to coarsen the resolution of the landslide observations to match that of the classifiers. In this case, we chose to set the threshold higher than the 1% threshold used in Figures 3 and 4 and considered only cells with >25% landslide area to be 'landslide affected' for the purpose of ROC and precision-recall analysis. Pixels containing more landslides are more important to identify, for example because the chance that a road within the pixel has been blocked by a landslide is higher. But if the threshold is set too high, not enough pixels are classed as 'landslide' to be able to carry out meaningful ROC or precision-recall analysis, especially in Lombok where the landslide density was lowest, and many regions of landsliding will be missed by the classifier . The choice of 25% is therefore somewhat arbitrary but represents a trade-off between reliability of the ROC and P-R analysis and sensitivity to pixels less affected by landslides.

We have also tested at other thresholds (1%, 10% and 50%). These results are now included as an additional figure in Supplementary Information and referred to in the main text. In general, more severely impacted pixels have a stronger signal in the coherence surfaces and so are easier to detect. Therefore, increasing this threshold (e.g. to 50%) produces a higher ROC AUC. The difference in ROC AUC between setting this threshold at 1% and 25% can be up to 0.18, but in most cases was less than 0.1. Again, this choice makes little difference to the relative ranking of the methods and thus does not influence our conclusions.

**Manuscript Change:**

*"We explore the effect that this choice of a 25% threshold to define a landslide pixel has on our results in the supplementary material, but varying this threshold between 1% and 50% was found to have little effect on the relative performance of different classifiers."* added at line 264 of the revised manuscript.

*"Finally, here we assessed classifier performance using ROC analysis, which does not require a threshold to be applied to the classification surface. However, if SAR methods are to be applied to future events for emergency response, it will be necessary to set a threshold between 'landslide' and 'non-landslide'. In this study the time between image acquisitions varied significantly between events, making it unlikely that a threshold could be selected that would work well for both Sentinel-1 and ALOS-2 across all events. However, this may be possible in the future when more events have been studied and SAR data with more regular acquisitions are available. Further work is therefore needed to establish such thresholds, which will be determined according to the requirements of emergency responders and their relative tolerance for false positives and false negatives"* added at line 510 of the revised manuscript.

**Comment 4**

'ARIA' method naming:

The use of the descriptor 'ARIA' method isn't standard to my knowledge, and may be misleading. JPL's ARIA Project does employ this method, although it is also working on many alternative approaches. Furthermore, the method, or similar variants of it, has been used by many groups (e.g. Fieldling et al (2005)) and is not referred to as the 'ARIA' method in these publications. One term I have seen applied is 'Coherence Change Detection' (e.g. Washaya et al. 2018), however this may not be the most useful, as all of your methods are change detection using coherence. Perhaps 'coherence loss' would be a better term?

Citations for the paragraph above:

- Fielding, E. J. et al. (2005) 'Surface ruptures and building damage of the 2003Bam, Iran, earthquake mapped by satellite synthetic aperture radar interferometric correlation', Journal of Geophysical Research: Solid Earth, 110(3), pp. 1–15. doi:10.1029/2004JB003299.

- Washaya, P., Balz, T. and Mohamadi, B. (2018) 'Coherence Change-Detection with Sentinel-1 for Natural and Anthropogenic Disaster Monitoring in Urban Areas', Remote Sensing, 10(7), p. 1026. doi: 10.3390/rs10071026.

**Response:** Thank you for drawing our attention to these papers, we will add these to the text and include a more complete description of damage-detection methods based on coherence loss. To match the naming of the method we refer to as "post-event coherence increase" (PECI), we will alter the text and refer to the "ARIA" method as "co-event coherence loss" (CECL)

**Manuscript changes:** the acronym *"ARIA"* has been altered to *"CECL"* throughout the manuscript, and the method renamed to "Co-event coherence loss".

*"This method was developed by the NASA Advanced Rapid Imaging and Analysis (ARIA) project for use in urban damage mapping… "* at line 164 of the original manuscript has been changed to *"The coherence loss between a pre-event interferogram and a co-event interferogram can be used to detect physical changes to the ground surface associated with an earthquake, such as surface rupture, building damage and landslides (Fielding et al., 2005; Washaya et al., 2018; Yun et al., 2015). This method has been applied by the NASA Advanced Rapid Imaging and Analysis (ARIA) project for use in urban damage mapping…"* at line 179 of the revised manuscript. References for Washaya et al. (2018) and Yun et al. (2015) have been added to the bibliography.

A reference to Fielding et al. (2005) was added at line 46 of the revised manuscript.

**Comment 5**

Line 73 - ï z£'The signal-to-noise ratio of each pixel in an InSAR image is described by its coherence'. This statement feels incomplete to meǎˇTin your work the 'signal' is the decorrelation of the pixel, and in circumstances where the phase is the signal (e.g. deformation time series), the decorrelation is only one contributor to the noise (e.g. can also have atmospheric and ionospheric noise which doesn't affect the coherence).

**Response:** By 'signal to noise', we meant the signal quality of the interferogram when it is used for ground deformation studies, not the signal in coherence studies, and were referring to high-frequency noise, not the longer wavelength nuisance signals that arise from ionospheric or tropospheric phase changes. We have reworded this statement to make it clearer.

**Manuscript Change:** *"The signal-to-noise ratio of each pixel in an InSAR image is described by its coherence. The coherence of a pixel can be estimated from the similarity between two SAR images in amplitude and phase difference, for a small ensemble of pixels n (Eq. 1, Just and Bamler, 1994):"* at line 72 of the original manuscript has been changed to "*When using an interferogram to map ground deformation, it is important that the signals recorded at a given location in the two SAR images are correlated, as decorrelation will result in high-frequency noise. In order to assess this and to identify noisy pixels, the coherence is estimated for every pixel from the similarity in the two SAR images in amplitude and phase difference, for a small ensemble of n pixels (Eq. 1, Just and Bamler, 1994):"* at line 75 of the revised manuscript.

**Comment 6**

Line 77 - I think some mention of the assumptions of the boxcar and sibling methods would be useful. My understanding is that the box-car method assumes that the surrounding pixels are statistically similar, and the sibling method identifies pixels that are similar through time (based on the amplitude) and assumes that these remain similar for subsequent acquisitions. I note that the sibling method is briefly explained on line 212, but more information in the introduction I feel would be helpful to the reader.

**Response:** Yes this is correct, we have added more information here for clarity

**Manuscript Change**: "*The ensemble of pixels used in the calculation can be selected either using a 'boxcar' method, in which a small rectangular ensemble of pixels immediately adjacent to and centred on the target pixel are used, or a 'sibling' method (e.g. Spaans and Hooper, 2016) where the ensemble of pixels is selected from within a wider window*" at line 77 of the original manuscript changed to "*The ensemble is chosen so that the pixels used in the calculation are expected to be similar. In a "boxcar" method, it is assumed that pixels immediately adjacent to and centred on the target pixel are similar to it (e.g. Hanssen, 2001; Yun et al., 2015). In a "sibling" method an assessment is carried out for every pixel to identify pixels that are statistically similar to it. For example, the sibling method of Spaans and Hooper (2016) identifies pixels that have similar amplitude behaviour through time.*" at line 81 of the revised manuscript.

**Comment 7**

perhaps include a definition of perpendicular baseline, and mention that orbital controls on modern satellites have rendered this a much smaller issue?

**Response:** Thank you, we have added this to the text.

**Manuscript Change:** *"When the perpendicular baseline of the SAR image pair used to form an interferogram is sufficiently small, this spatial component will be small compared to any temporal decorrelation (Zebker and Villasenor, 1992)"* at line 87 of the original manuscript changed to *"This decorrelation is particularly sensitive to the SAR image pair's perpendicular baseline (the distance between the locations at which the satellite acquired the two SAR images measured perpendicular to both the flight and look directions). When the perpendicular baseline of the image pair used to form an interferogram is sufficiently small, this spatial component will be small compared to any temporal decorrelation (Zebker and Villasenor, 1992). For modern satellites, this will be the case most of the time."* at line 93 of the revised manuscript.

**Comment 8**

Line 100 - add citation on the effect of vegetation on coherence at different wavelengths?

**Response:** We have cited Zebker and Villasenor (1992) here, who compare decorrelation due to movement of scatterers at L-band and C-band wavelengths.

**Manuscript Change:** Reference to Zebker and Villasenor added at line 108 of the revised manuscript

**Comment 9**

Line 106 'as it is impossible to combine data from both tracks' - it would be helpful to clarify what is meant by 'combine' (i.e. you can't calculate the coherence between different tracks)

**Manuscript Change:** "*As it is impossible to combine data from both tracks*" at line 106 of the original manuscript has been changed to "*As it is impossible to calculate a combined coherence surface using data from two tracks*" at line 113 of the revised manuscript

**Comment 10**

Line 119 - what is meant by 'high resolution' data for ALOS-2? Might be helpful to mention the acquisition mode for clarity?

**Response:** We have specified in the text that these data were acquired in "stripmap" mode at a resolution of 3 – 10 m. The resolutions of each track can be found in Table 1.

**Manuscript Change:** "*(acquired in stripmap mode at a resolution of 3 - 10 m*)" added at line 127 of the revised manuscript.

**Comment 11**

Line 165 - need to clarify that the user must choose a coherence loss threshold for flagging damage, the current sentence structure implies that all pixels that have any coherence decrease at all are flagged as damaged

**Response** Here we apply this method to landslides and do not apply a threshold for the ROC analysis, but in the original Yun et al. 2015 paper, they use 0 as the threshold, so that any pixel whose coherence decreases is flagged as damaged. We have clarified this in the text.

**Manuscript Change:** "*The pre-event surface is then subtracted from the co-event surface (Yun et al., 2015).*" at line 167 of the original manuscript changed to "*The pre-event surface is then subtracted from the co-event surface and pixels whose coherence has decreased are flagged as damaged (Yun et al., 2015*)." at line 187 of the revised manuscript

**Comment 12**

Line 179 - I think it's worth reminding the reader here that the sibling pixels have been specifically identified to be behaving similarly before the earthquake by looking at the amplitude time series, and it's on that basis that they're expected to behave similarly

**Response** Yes, that's a good suggestion, we have added this to the text.

**Manuscript Change:** "*When a sibling-based method such as that of Spaans and Hooper (2016) is used to estimate coherence, the coherence of a pixel becomes dependent on 'siblings' that are not immediately adjacent to it but that are expected to behave similarly.*" at line 177 of the original manuscript changed to "*When a sibling-based method is used to estimate coherence, the coherence of a pixel becomes dependent on "siblings" that are not immediately adjacent to it but that are expected to behave similarly. In the method of Spaans and Hooper (2016) used here, an ensemble of siblings is selected for every pixel that have similar amplitude behaviour in a time series of pre-event imagery*" at line 197 of revised manuscript.

**Comment 11**

Line 185 - post-event coherence - make it clear that this coherence is calculated using the boxcar method (which I assume it is?)

**Response:** This was stated at line 176, but we have altered the text to ensure it is clear that the boxcar coherence is used in sections 3.3.1, 3.3.3, 3.3.4, and 3.3.5.

**Manuscript Change** "*A "boxcar" coherence estimate is used for CECL, PECI, ΔC_sum and ΔC_max.*" added at line 176 of the revised manuscript.

**Comment 12**

Line 233 - 'If over 95% of an aggregate pixel is made up of masked pixels, the aggregate pixel was masked' - how do you choose this number? Are you results sensitive to this choice? What fraction of pixels end up being masked out?

**Response:** The effect of this choice on ROC AUC is very low (< 0.02). However, altering this can significantly alter the number of pixels masked, for example on track 19 in Nepal, altering this threshold from 95% to 5% decreases the number of aggregate pixels used in the analysis from 246,375 to 148,829. The effect was much weaker in Hokkaido and Lombok, where fewer pixels were masked due to distortion on steep slopes. For example, in Hokkaido on track 68, altering the threshold from 95% to 5% decreases the number of aggregate

pixels from 19,754 to 19,235. We chose to set the threshold high to avoid masking any landslides unnecessarily and to therefore maintain good spatial coverage for our classification surface.

**Manuscript change***: "This high threshold of 95% was chosen to minimise the loss of spatial coverage due to the masks. Varying the threshold between 95% and 5% had little difference in terms of the number of pixels used in the analysis in Hokkaido and Lombok (<5%), but in Nepal, where more pixels were masked due to distortion on steep slopes, decreasing the threshold to 5% resulted in a loss of coverage of around 40% on S085a. Altering this threshold made very little difference to the results presented in Section 4.1.*
*."* added at line 254 of the revised manuscript.

**Comment 13**

Line 234 - ï z£'In this study, we did not attempt to map SAR classification surface values directly to landslide areal density values, as this has not been attempted in previous studies and may not be possible' - would be good to get more info about why this is/ isn't possible, and a citation for the previous work. Is this comment based on the work of Burrows et al. (2019)?

**Response:** All of the studies done so far applying SAR data to landslide detection have used a binary validation landslide dataset and have not attempted to recreate landslide areal density (Aimaiti et al. 2019; Burrows et al. 2019; Ge et al. 2020; Jung and Yun 2019; Masato et al. 2020; Yun et al. 2015). Although it may be possible to extract landslide areal density from SAR data, this is uncertain as differences in viewing geometry, land cover type and (particularly with the ALOS-2 data used here) differences in the temporal baselines of the coherence maps may mean that there is too much variation between events to establish rules on this. This is now clarified in the text.

**Manuscript change: "***In this study, we did not attempt to map SAR classification surface values directly to landslide areal density values, as this has not been attempted in previous studies and may not be possible*" at line 234 of the original manuscript changed to *"In this study, we did not attempt to map SAR classification surface values directly to landslide areal density values, as this has not been attempted in previous studies (e.g. Aimaiti et al., 2019; Burrows et al., 2019; Jung and Yun, 2019; Yun et al., 2015) and may not be possible due to differences in viewing geometry, land cover and, particularly with the ALOS-2 data used here, differences in temporal baseline between events."* at line 259 of the revised manuscript.

**Comment 14**

Line 236 - how did you select the 25% threshold? How much does this choice affect your results? How large is this area, and what fraction of the total landslide area is missed by virtue of the fact that it falls below this threshold? Would be helpful to have these questions discussed in the text, particularly if this choice has a large affect on AUC values

**Response:** The choice of this threshold has been explained in our response to Comment 3, Reviewer 2**.** The effect of this choice is now demonstrated in Supplementary material where we varied this threshold between 1% and 50%. In general, setting the threshold high results in slightly higher ROC AUC because the signal of an aggregate pixel containing more landslide pixels is stronger but the relative performance of different classifiers remains the same.

For a 200 x 220 m pixel, 25% corresponds to an area of 11,000 m2 out of the total 44,000 m2 area of each aggregate pixel. Applying this threshold results in around 27% of the total landslide area being classified as 'non-landslide' in Hokkaido, around 61% in Nepal and in Lombok, where the average landslide size is comparably small, around 86% for Lombok-1 and 85% for Lombok-2. When the threshold is dropped to 10%, the total excluded landslide area decreases to 6% in Hokkaido, 26% in Nepal and 55% and 53% in Lombok-1 and -2 respectively. Information on the proportion of excluded pixels with the threshold set at 1%, 10%, 25% and 50% is now included along with the supplementary information supplied in response to Comment 3, Reviewer 2

**Manuscript Change** See comment 3, reviewer 2

**Comment 15**

Line 277 - 'ï z£the 6-day Sentinel-1 acquisition window' - this is a little unclear, to my mind 'window' implies that data was acquired at some point in that time range. Rather than 'window' could say 'acquisition frequency'?

**Response** We have changed this to repeat time to avoid confusion.

**Manuscript change:** "*Acquisition window*" at line 277 of the original manuscript changed to *"acquisition repeat time"* at line 313 of the revised manuscript.

**Comment 16**

Line 287 - for clarity it may be worth reminding the reader that the L-band image is from the ALOS 2 satellite

**Manuscript Change:** *"from ALOS-2"* added at line 323 of the revised manuscript.

**Comment 17**

Line 324 - a brief discussion of what is meant by 'higher quality siblings' would be helpful (I imagine that they're more statistically similar?)

**Response:** If siblings are selected using SAR images acquired more recently before the earthquake, then there is less time for pixels to have been altered by changes to the ground surface, and their behaviour is more likely to remain similar in the co-seismic image (assuming there are no landslides)

**Manuscript change**

*"As Sentinel-1 imagery is acquired every 12 days, more images were available for this calculation and were acquired over a shorter time period (Fig. 1). Therefore it may be the case that the siblings selected by RapidSAR when using Sentinel-1 imagery were of a higher quality than when using ALOS-2, giving a more reliable sibling coherence estimate."* in the original manuscript changed to *"As Sentinel-1 imagery is acquired every 12 days, more images were available for this calculation and were acquired over a shorter time period. This allows less time for pixels to be altered by changes to the ground surface, meaning that, for non-landslide pixels, a pixel and its siblings are likely to be more similar. In this way, the siblings selected by RapidSAR for Sentinel-1 imagery may have been of a higher quality than those for ALOS-2, giving a more reliable coherence estimate."* at line 357 of the revised manuscript.

**Comment 18**

Line 425 - You mention rivers giving false positives for the Bx-S method. Would it be possible to identify rivers by applying the Bx-S method to a pre-seismic coherence image (using the same pixel siblings) and then use that as a mask on your co-seismic landslide image? You could propose this if so.

**Response:** Yes this might be possible and could be a good way to remove these false positives. This has been covered in our response to Reviewer 1, Comment 15

**Manuscript Change:** See Reviewer 1, Comment 15

**Comment 19**

Line 469 - ï z£ 'The ARIA method is the best performing method when only one L-band image is available' - does this mean 'one post-seismic L-band SAR image'? I think this could be clarified, as I could read this to mean that only on L-band SAR image is available in total.

**Response** Thank you for pointing this out, we have changed this for clarity

**Manuscript Change:** *"only one L-band image"* at line 469 of the original manuscript changed *to "only one post-event L-band image"* at line 521 of the revised manuscript.

**Comment 20**

Figure 6 - how is the threshold for these plots chosen? Could you clarify in the caption?

**Response:** The 10% of pixels most likely to be landslides are plotted. This has been added to the figure caption

**Technical Corrections**

4 technical corrections were suggested by the reviewer, all of which have been implemented in the manuscript.

1. Technical Corrections: For words in quotation marks apostrophes are being used throughout, rather than opening and closing quotation marks. Please adjust.
2. Line 185 - double use of 'by a landslide' in this sentence.
3. Line 235 and 443 - add a comma after 'Thus' (consistent with 'Thus' on line 10)
4. Table 1 - 'x' has inconsistent formatting or is missing in some cases

[revised manuscript text omitted]